# Chloride channels regulate differentiation and barrier functions of the mammalian airway

Mu He[1][†]*, Bing Wu[2][†], Wenlei Ye[1], Daniel D Le[2], Adriane W Sinclair[3,4], Valeria Padovano[5], Yuzhang Chen[6], Ke-Xin Li[1], Rene Sit[2], Michelle Tan[2], Michael J Caplan[5], Norma Neff[2], Yuh Nung Jan[1,7,8], Spyros Darmanis[2]*, Lily Yeh Jan[1,7,8]*

[1]Department of Physiology, University of California, San Francisco, San Francisco, United States; [2]Chan Zuckerberg Biohub, San Francisco, United States; [3]Department of Urology, University of California, San Francisco, San Francisco, United States; [4]Division of Pediatric Urology, University of California, San Francisco, Benioff Children's Hospital, San Francisco, United States; [5]Department of Cellular and Molecular Physiology, Yale University School of Medicine, New Heaven, United States; [6]Department of Anesthesia and Perioperative Care, University of California, San Francisco, San Francisco, United States; [7]Department of Biochemistry and Biophysics, University of California, San Francisco, San Francisco, United States; [8]Howard Hughes Medical Institute, University of California, San Francisco, San Francisco, United States

*For correspondence:
mu.he@ucsf.edu (MH);
spyros.darmanis@czbiohub.org
(SD);
Lily.Jan@ucsf.edu (LYJ)

[†]These authors contributed equally to this work

Competing interests: The authors declare that no competing interests exist.

**Abstract** The conducting airway forms a protective mucosal barrier and is the primary target of airway disorders. The molecular events required for the formation and function of the airway mucosal barrier, as well as the mechanisms by which barrier dysfunction leads to early onset airway diseases, remain unclear. In this study, we systematically characterized the developmental landscape of the mouse airway using single-cell RNA sequencing and identified remarkably conserved cellular programs operating during human fetal development. We demonstrated that in mouse, genetic inactivation of chloride channel *Ano1/Tmem16a* compromises airway barrier function, results in early signs of inflammation, and alters the airway cellular landscape by depleting epithelial progenitors. Mouse *Ano1*[-/-] mutants exhibited mucus obstruction and abnormal mucociliary clearance that resemble the airway defects associated with cystic fibrosis. The data reveal critical and non-redundant roles for *Ano1* in organogenesis, and show that chloride channels are essential for mammalian airway formation and function.

## Introduction

The highly conserved respiratory system of air breathing animals represents a major interface between internal organs and the outer environment. In the course of a typical human lifespan, approximately 200 to 400 million liters of air are conducted via the respiratory system (*Ganesan et al., 2013*; *Rackley and Stripp, 2012*). While airway function has been adapted for organismal physiology and aging (*Sharma and Goodwin, 2006*), it remains vulnerable to deleterious genetic and environmental factors. Cystic fibrosis (CF), which primarily targets the respiratory system, is one of the most common recessively inherited disorder caused by the deficient *CFTR* gene that encodes a chloride channel (*Stoltz et al., 2015*). The main features of CF airway diseases include mucus obstruction and repetitive infections and inflammation, which often lead to severe

airway remodeling and respiratory failure (*Regamey et al., 2011*). It has been reported that CF symptoms emerge as early as the fetal stage, indicating that alterations of airway development can have a profound impact on the respiratory function later in life (*Gosden and Gosden, 1984*; *Larson and Cohen, 2005*; *Regamey et al., 2011*; *Verhaeghe et al., 2007*).

Mouse mutants that lack *Cftr* do not exhibit airway defects similar to those found in cystic fibrosis patients (*Lavelle et al., 2016*; *McCarron et al., 2018*). This led to the hypothesis that chloride channels may play species-specific roles and that other chloride channels, such as calcium-activated chloride channel (CaCC), may compensate for the lack of *Cftr* in mice (*Clarke et al., 1994*). *Ano1*, also known as *Tmem16a*, is a CaCC in the Anoctamin/TMEM16 family. *Ano1* regulates intracellular chloride homeostasis (*He et al., 2017*) and is required for survival (*Lek et al., 2016*; *Rock et al., 2008*), and mouse mutants that lack *Ano1* exhibit abnormal trachea morphology (*Rock et al., 2008*; *Rock et al., 2009*). Given its function as a chloride channel in the airway, ANO1 is a candidate drug target in the modulation and management of CF (*Amaral and Beekman, 2020*). Despite many efforts to identify agonists and activators for the chloride channel ANO1, the physiological role for ANO1 in the airway development and regeneration remain unclear.

To systematically and unbiasedly characterize the cellular processes important for airway development, as well as to define cellular origins of disease phenotypes that depend on chloride channels, we used single-cell RNA sequencing technology (scRNAseq) to profile mouse embryonic and neonatal trachea as well as human fetal trachea. We uncovered conserved cell types implicated in monogenic and complex-trait airway diseases and defined cell states associated with epithelial cell differentiation. In parallel, we analyzed the developmental landscape of the mouse trachea in the absence of *Ano1*. Loss of *Ano1* compromises airway barrier function, results in early signs of inflammation, and alters the airway cellular landscape by depleting epithelial progenitors. The data reveal critical and non-redundant roles for *Ano1* in organogenesis, and show that chloride channels are essential for mammalian airway formation and function. Because *Ano1* and *CFTR* are expressed in orthologous cell types of mouse and human airway epithelium, respectively, our work provides a tractable animal model for understanding the roles of chloride channels in human airway development and pathogenesis.

## Results

### Inactivation of *Ano1* chloride channel compromises airway functions

Mucus accumulation, a hallmark of many chronic airway diseases, has been previously reported in the newborn airway of $Ano1^{-/-}$ knockout mice (*Rock et al., 2009*). To explore the cellular origin of mucus obstruction, we first determined whether removal of *Ano1* led to any alternation in mucus producing cells in a $Ano1^{-/-}$ knockout mouse line (*Rock et al., 2008*). Using fluorescently labeled Jacalin, a plant-based lectin that recognizes airway glycoproteins and mucin components (*Ostedgaard et al., 2017*), as well as antibody against SCGB1A1, a low-molecular-weight protein enriched in airway secretory cells, we observed a massive expansion of the secretory cell population in $Ano1^{-/-}$ knockout airway at postnatal day 0 (P0) (*Figure 1A*; *Figure 1—figure supplement 1A*). In addition, Jacalin-positive mucus substance was observed in $Ano1^{-/-}$ knockout airway lumen (*Figure 1A*). At P3 and P5, Periodic Acid–Schiff stain (PAS) and Alcian Blue staining of airway histological sections consistently demonstrated strong mucus obstruction of the respiratory tract and alveolar simplification in $Ano1^{-/-}$ neonatal lung (*Figure 1B*; *Figure 1—figure supplement 1B,C,D,E*).

We next assessed tissue-level mucociliary clearance by characterizing the flow dynamics of fluorescent beads generated by airway motile cilia. We dissected wild-type and $Ano1^{-/-}$ mutant trachea at P2 to P3 and spliced the trachea open along the dorsal smooth muscles to expose the airway lumen. We then placed individual trachea sample in imaging media with fluorescent beads and visualized the movement of beads via confocal live imaging. At P2, wild-type trachea showed directional flow at $0.90 \pm 0.22$ µm/s from the distal to the proximal trachea (*Figure 1—video 1*). In contrast, Ano1 mutants showed minimal and sometimes reversed flow at $0.42 \pm 0.18$ µm/s (*Figure 1C,D*; *Figure 1—video 2*), significantly slower than the velocity observed in littermate controls.

We next quantified the secretory cells and ciliated cells, presented as a percentage of the total airway luminal cells, by using conventional cell type markers. Immunostaining of SCGB1A1 showed a significant increase in the number of secretory cells in the mutant airway compared to wild-type

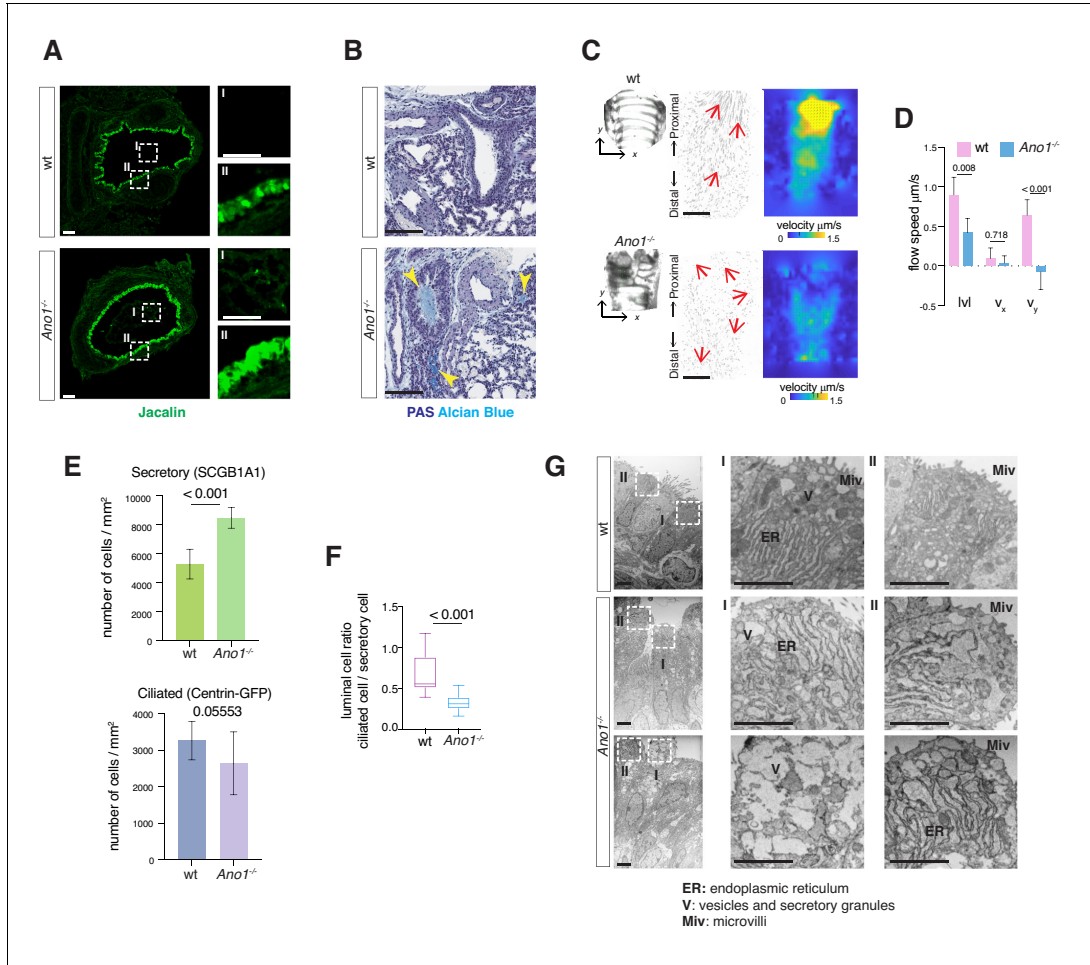

**Figure 1.** Mucus cell hyperplasia in *Ano1* null mutants. (A) Jacalin-Alexa488 (green) labeling of glycoprotein-producing cells and mucin components in wild-type and *Ano1*[-/-] mutant trachea at P0. Inset I shows the tracheal lumen; mucosubstances were apparent in the mutant trachea lumen. Inset II shows glycoprotein-producing cells. Scale bar corresponds to 50 μm. n = 4 for each genotype examined. (B) PAS and Alcian blue staining of airway and mucus in wild-type and *Ano1*[-/-] mutant trachea at P3. Yellow arrowheads indicate mucus accumulation in the mutants. Scale bar corresponds to 100 μm. n = 4 for each genotype examined. See *Figure 1—figure supplement 1* for additional analysis for pulmonary defects associated with *Ano1*[-/-] mutants. (C) Differential interference contrast (DIC) images of flat-mounted trachea, flow path lines, and velocity magnitude of ciliary flow generated by wild-type and *Ano1*[-/-] mutant trachea samples at P2. Flow directions are indicated by arrows in red. X represents the medial-lateral axis, while Y represents the anterior-posterior axis. (D) Velocities of ciliary flow at P2. Wild-type trachea showed directional flow at 0.90 ± 0.22 μm/s from the distal to the proximal trachea (*Figure 1—video 1*). Mutants showed minimal and sometimes reversed flow at a lower speed of 0.42 ± 0.18 μm/s (*Figure 1—video 2*). n = 3 for each genotype. p values are indicated (multiple t-test). Error bars represent standard deviation (S.D.). (E) Quantification of SCGB1A1+ secretory cells and Centrin-GFP+ ciliated cells of wild-type and *Ano1*[-/-] mutant trachea samples at P3. n = 5 for each genotype. p-value (unpaired t-test) are indicated. Error bars represent S.D. (F) Ratio of ciliated cells over secretory cells at P3. n = 5 for each genotype. p value is indicated (unpaired two-tailed t-test). Box and whisker plot shows 10–90 percentile. (G) TEM images of wild-type and *Ano1*[-/-] mutant tracheal epithelial cells. Mutant secretory cells show reduced microvilli (Miv) and abnormal intracellular organizations, including dilated ER lumen (ER) and accumulation of vesicles (V). Scale bars indicate 2.5 μm. n = 3 for each genotype.

The online version of this article includes the following video and figure supplement(s) for figure 1:

**Figure supplement 1.** Additional mucus and alveolar defects associated with *Ano1* knock-outs.

**Figure 1—video 1.** Flow Movie made from wild-type trachea.

https://elifesciences.org/articles/53085#fig1video1

**Figure 1—video 2.** Flow Movie made from Ano1[-/-] mutant trachea.

https://elifesciences.org/articles/53085#fig1video2

(*Figure 1E,F*; *Figure 1—figure supplement 1A*). To assess the number of ciliated cells, we crossed *Ano1*[+/-] heterozygous animals to a reporter mouse line Arl13b-mCherry/Centrin-2-GFP and generated *Ano1*[-/-] homozygous mutants that express the fluorescent reporters. Based on the expression of Centrin2-GFP, the numbers for ciliated cells were unaffected in *Ano1*[-/-] mutant airway compared to littermate wild-type (*Figure 1E*; *Figure 1—figure supplement 1A*). Transmission electron microscopy (TEM) analysis revealed subcellular defects in mutant neonatal epithelial cells, including short microvilli, dilated ER lumen and accumulation of large amorphous vesicles (*Figure 1G*).

## A cellular atlas of the developing airway in mice and humans

To comprehensively characterize the cellular origins of these complex airway defects, we carried out scRNAseq of mouse trachea to establish an atlas of 16,000 wild-type cells from embryonic day 15 (E15), E16, P1, and P4 (*Figure 2A,B*; *Figure 2—figure supplement 1A,B*). In parallel, we analyzed a matching number of transcriptomes from *Ano1* mutant tracheal cells captured at E16, P1 and P4. Our wild-type single-cell embryonic and neonatal atlas largely agrees with previously published atlases of the adult airway in the expression of marker genes for major epithelial cell types (*Figure 2B*; *Figure 2—figure supplement 1C*; *Montoro et al., 2018*; *Plasschaert et al., 2018*). In addition, our data include previously uncharacterized fibroblasts, vascular cells, chondrocytes, airway smooth muscle cells, immune cells, and neuronal cell types (*Figure 2B*; *Figure 2—figure supplement 1D,E,F*; *Figure 2—source data 1*). We also profiled approximately 9600 human fetal trachea cells at gestation weeks 21 and 23 (GW21 and GW23) (*Figure 2C*). This allowed us to identify orthologous cell types and cell states defined by molecular markers similar to those we observed in the mouse airway, and to uncover cell types specific to the human airway, such as cell types associated with the submucosal glands in humans (*Figure 2D,E*). Taken together, we observed that major cell types of the developing airway and molecular markers associated with each cell type are highly conserved between mice and humans (*Figure 2E*).

## The developmental landscape of the mouse airway epithelial cells

The temporal dimension of our data allowed us to examine the molecular programs operating during mammalian airway differentiation. In wild-type mice, coordinated differentiation programs of luminal cell types, including ciliated and secretory cells, were initiated between E15 and E16 (*Figure 3A*). *Spdef* and *Creb3l1*, two transcription factors that promote the secretory cell fate (*Chen et al., 2009*; *Gregorieff et al., 2009*) and *Foxj1*, a transcription factor essential for the motile ciliated cell formation (*You et al., 2004*; *Yu et al., 2008*), were upregulated at E16 compared to E15 (*Figure 3B*). Conversely, the number of cycling cells, reflected by a calculated cell cycle score, was diminished as differentiated cells began to emerge (*Figure 3C*; *Figure 3—source data 1*). At E15, the broad presence of *Trp63* but low level of *Krt5*, and the marked expression of *Id2*, *Id3*, *Wnt7b*, and *Cldn6* indicate that undifferentiated cells dominate the *Epcam*[+] epithelium (*Figure 3D*). When luminal cells begin to emerge at E16, *Trp63*[+] basal cells switch to a different expression program, consisting of *Krt5*, *Krt15*, *Aqp3*, and *Aqp4*, which may play important roles in the epithelial barrier function (*Kreda et al., 2001*; *Figure 3D*).

At E16, in addition to *Spdef* and *Creb3l1*, the immature secretory program showed a pronounced expression of *Cited1*, a transcription cofactor that activates TGF-β and BMP signals (*Plisov et al., 2005*). Conversely, the neonatal secretory program was distinguished by elevated expression of *Gp2* and *Tff2*, which are important mucosal proteins and markers for goblets cells found in the adult airway (*Hase et al., 2009*; *Montoro et al., 2018*; *Nikolaidis et al., 2006*; *Wills-Karp et al., 2012*), as well as *Sftpd*, which encodes a surfactant protein and is involved in the innate immune program (*Brandt et al., 2008*; *Mackay et al., 2016*; *Figure 3D*). The immature and mature secretory cell transcriptomes exhibited non-overlapping gene modules characteristic of distinct biological functions. The immature secretory program includes gene modules required for protein folding and trafficking, which are critical for the secretory pathway (*Trombetta and Parodi, 2003*), while the mature secretory program features processes involved in the regulation of cell division (*Figure 3E*).

Ciliated cells, which form motile cilia, are essential for mucus clearance. In our dataset, both embryonic and postnatal ciliated cells expressed *Foxj1* (*Figure 3D*). Immature ciliated cells, which are predominantly present during embryogenesis, were characterized by *Foxn4*, *Ccna1*, *Ccno*, *Mcidas*, as well as by uncharacterized markers, such as *Shisa8*, which is an auxiliary subunit for the

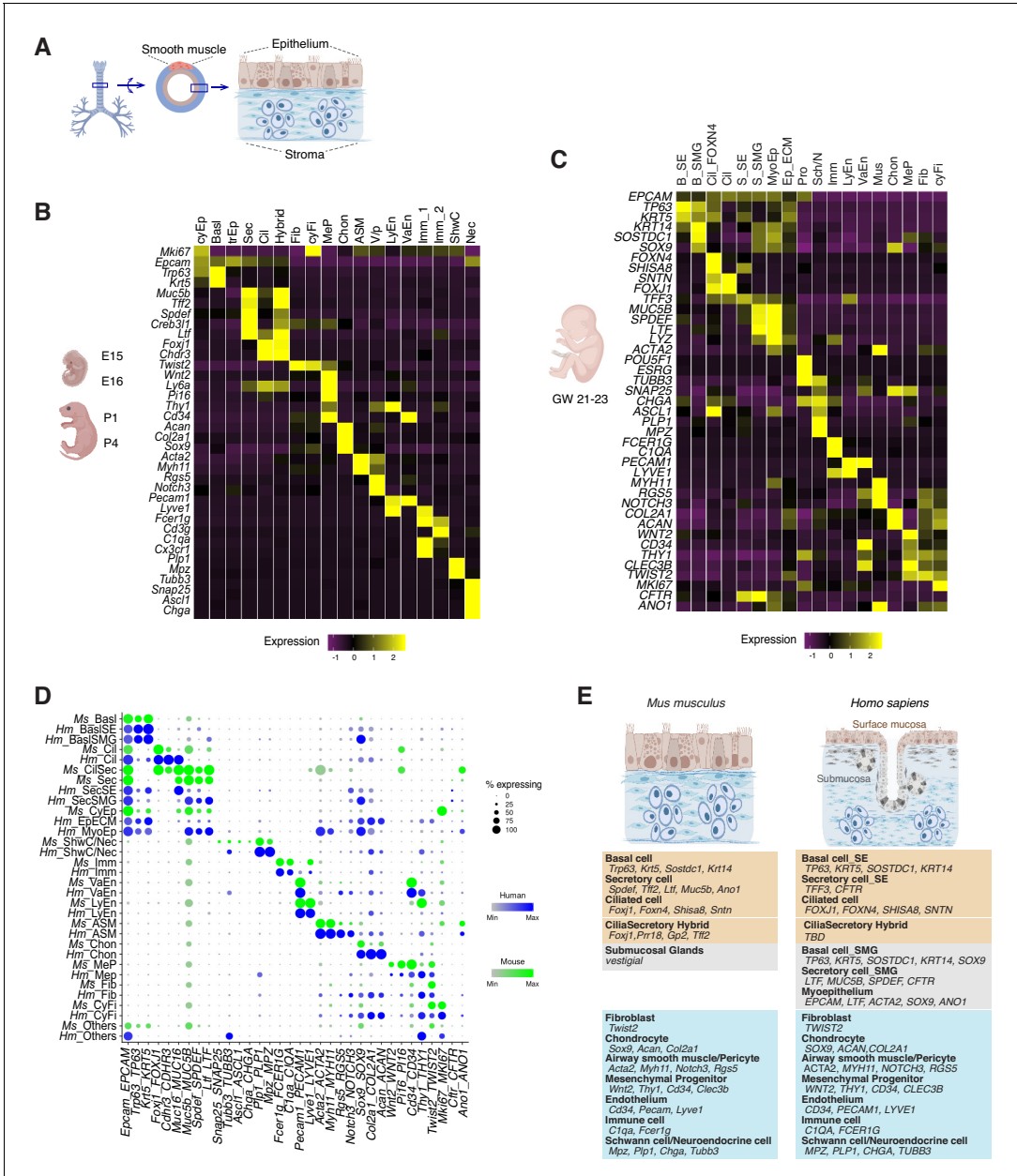

**Figure 2.** Transcriptomes of the developing mouse and human trachea. (**A**) Cartoons of the mammalian trachea anatomy. (**B**) Four developmental stages are included in this study: embryonic day 15 (E15), E16, postnatal day 1 (P1), and P4. Heat map illustrating the average expression levels of marker genes for each cell cluster identified at E15, E16, P1 and P4. Gene expression has been normalized, log-transformed, and z-score transformed. The proportion of cell types and cell states at different developmental stages are shown in *Figure 2—figure supplement 1*. A list for differentially expressed genes for all cell types is included in *Figure 2—source data 1*. (**C**) Heat map showing the average expression levels of cell identity marker genes for all cell types identified in human fetal trachea of gestation week (GW) 21–23. Gene expression has been normalized, log-transformed, and z-score transformed. (**D**) Dot plot depicting expression patterns of mouse and human orthologous cell types and states identified from this study. The size of the dot encodes the percentage of cells expressing the gene, while the color encodes the mean of expression level which has been normalized, log-transformed, and z-score transformed. Cell type legends for (**B**–**D**) Basl: basal cells; BaslSE: basal_surface epithelium; BaslSMG: basal_submucosal glands; Cil: ciliated cells; CilSec: cilia-secretory hybrid cells; Sec: secretory cells; SecSMG: secretory cells_submucosal gland; Hybrid: cilia-scretory hybrid cells; CyEp: cycling epithelium; EpECM: epithelium_ECM+ ; MyoEp: myoepithelium; Schw: Schwann cell precursors; Nec: neuronal cells; Imm: immune cells; VaEn: vascular endothelium; LyEn: lymphatic endothelium; ASM: airway smooth muscles: Chon: chondrocytes; MeP: mesenchymal progenitors; Fib: fibroblasts; CyFi: cycling fibroblasts; Pro: progenitor (human). (**E**) Summary of cell types and marker genes reflecting similarity and distinction between mouse embryonic and neonatal trachea and human fetal trachea.

The online version of this article includes the following source data and figure supplement(s) for figure 2:

*Figure 2 continued on next page*

*Figure 2 continued*

**Source data 1.** Gene lists for all cell types identified from the developing mouse airway.
**Figure supplement 1.** Construction of the developing mouse trachea atlas.

AMPA receptor and a member of the CKAMP glutamate receptor family required for synaptic trans-mission (*Farrow et al., 2015*). The immature ciliated program includes pathways involved in protein synthesis (*Figure 3F*). RNA FISH (*fluorescent in-situ hybridization*) analysis of *Mcidas* and *Shisa8* showed colocalizations with *Foxn4* and *Foxj1* in E16 wild-type trachea (*Figure 3—figure supplement 1A,B*). In contrast, postnatal mature ciliated cells upregulate *Sntn*, which encodes a phosphati-dylserine binding protein localizing to the tip of motile cilia (*Kubo et al., 2008*), and express various membrane receptors, such as *Cdhr3* and *Ldlrad1*, which are involved in rhinovirus infection (*Basnet et al., 2019*; *Figure 3D*). Overall, the mature ciliated program highlights molecular path-ways that are involved in axoneme assembly, apical junction organization, as well as cilia movement and motility (*Figure 3F*). An enrichment of these distinct gene modules reflects a structural and func-tional maturation of postnatal ciliated cells. The majority of these ciliated cell markers are conserved between mice and humans (*Figure 3—figure supplement 1C,D*).

## Identification of a cilia-secretory hybrid cell state in the neonatal airway

In the postnatal mouse dataset, we identified a population of epithelial cells expressing two gene modules, the ciliated-cell module and the secretory-cell one (*Figure 4A*). Using neonatal tracheal samples, we validated the presence of a cilia-secretory hybrid cell type by FISH analysis of *Foxj1*, *Gp2* and *Prr18*, a novel ciliated cell marker (*Figure 4B*). Based on immunofluorescence staining, these hybrid cells expressed FOXJ1 and a secretory cell marker SCGB1A1 (*Figure 4C*; *Rawlins et al., 2009*; *Zhang et al., 1997*). Transmission electron microscopy (TEM) revealed that a subset of luminal cells exhibit both characteristic cilia axoneme and intracellular vesicles, indicating that hybrid cells indeed possess two sets of machineries required for motility and secretion, respectively (*Figure 4D*).

To probe the developmental origin of these cilia-secretory hybrid cells, we assessed their pres-ence in a mouse mutant lacking *Pofut1*, which encodes an enzyme required for Notch ligand proc-essing (*Stahl et al., 2008*). Because the Notch pathway is essential for cell fate decision in the airway, *Pofut1*mutants fail to produce secretory cells and contain predominantly ciliated cells in the trachea (*Tsao et al., 2009*). Compared to littermate controls, abundant ciliated cells with motile cilia, marked by acetylated-α tubulin, were present in *Pofut1-/-* newborn mutants. *Pofut1-/-* mutants lacked secretory cells that are only expressing *Gp2*, but exhibited double positive *Foxj1*- and *Gp2*-expressing hybrid cells, indicating that hybrid cells are derived from a ciliated cell linage (*Figure 4—figure supplement 1A*). To further examine whether hybrid cells originate from a ciliated cell line-age, we performed an in vivo lineage-tracing approach. We crossed an inducible line *Foxj1-Cre^ERT2: GFP* to the *ROSAmT/mG* reporter line to trace *Foxj1+* lineages at the onset of airway differentiation. Because endogenous GFP flourescence signal of *Foxj1-CRE/ERT2-GFP* was very weak, we traced cells derived from the *Foxj1*+ lineage based on the expression of membrane-bound GFP. About 50% of *Foxj1Cre* labeled mGFP+ cells at this stage expressed cytoplasmic SCGB1A1 at P0 to P1, suggest-ing that a major proportion of the neonatal hybrid cells are likely derived from a ciliated cell lineage (*Figure 4—figure supplement 1B,C*).

## Mapping monogenic and complex trait disease-associated genes to cell types of the airway

We next evaluated various cell types from which complex airway disorders may arise. We examined the expression landscape of genetic risk loci associated with chronic obstructive pulmonary disease (COPD) and pulmonary fibrosis (*Sakornsakolpat et al., 2019*), as well as modifier loci associated with CF (*Corvol et al., 2015*; *Figure 4—figure supplement 1D*). Many of the airway disease associ-ated genes are expressed by cells of the airway epithelium, including modifier loci for CF severity, such as *Muc20* and *Ehf*, as well as risk genes shared between COPD and pulmonary fibrosis, such as *Fam13a* and *Dsp*. Overall, risk genes implicated in these airway diseases are expressed in various

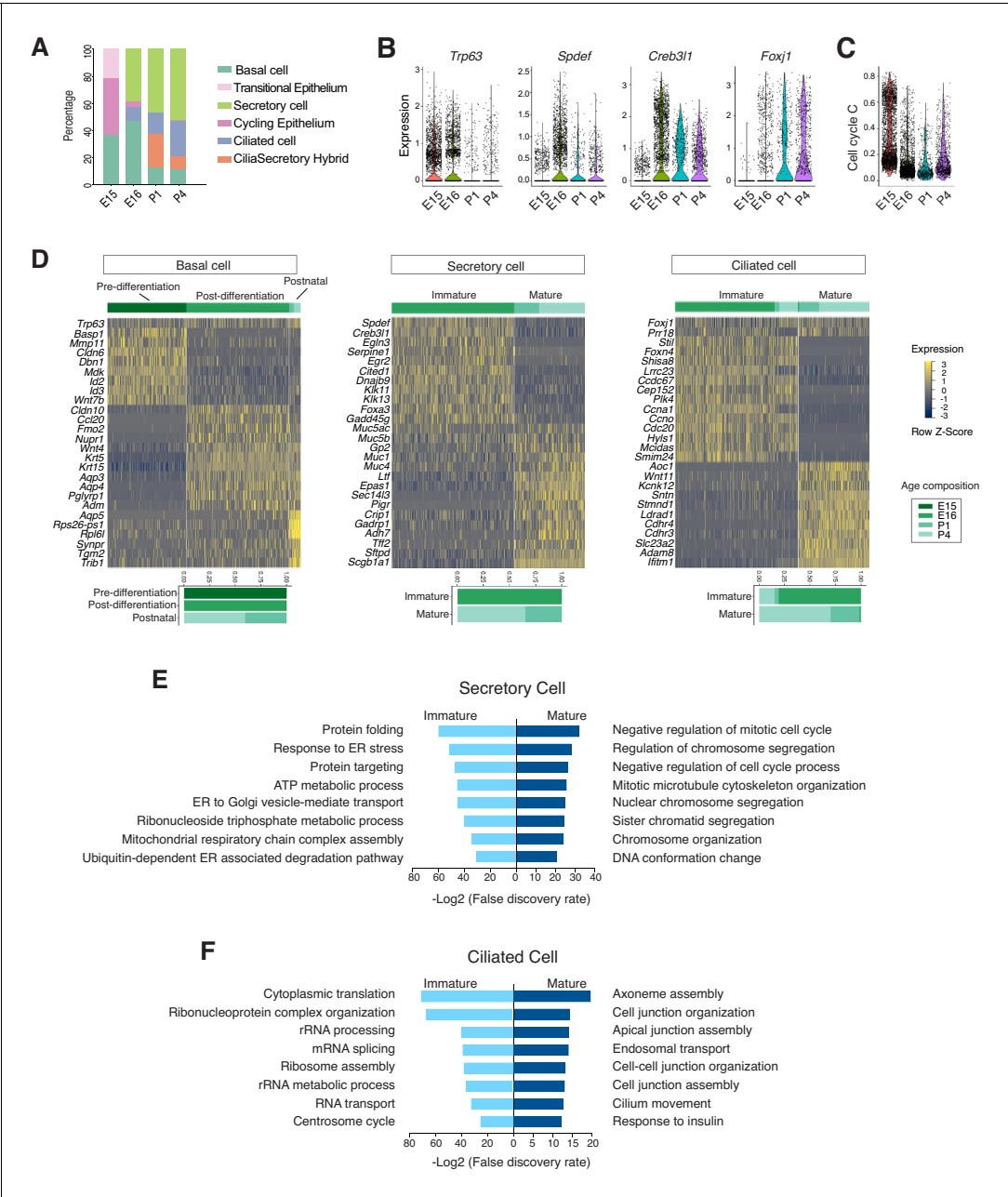

**Figure 3.** Developmental landscape of tracheal epithelial cells. (A) Horizontal bar graphs showing the cellular composition of mouse tracheal epithelial cells from different developmental and neonatal time points. (B) Violin plots showing the expression of *Trp63*, *Spdef*, *Creb3l1*, and *Foxj1* in epithelial cells at different developmental time points. Gene expression has been normalized and log-transformed. (C) scGPS showing cell-cycle gene expression in epithelial cells at different time points. scGPS (single-cell Geneset Percentile Score) is an expression enrichment analysis that utilizes sets of genes underlying certain biological and pathological processes to infer functional profiles for each cell. The cell cycle gene list can be found in *Figure 3— source data 1*. (D) Heat maps showing transcriptional profiles of tracheal basal, ciliated, and secretory cells across different developmental time points, with age compositions of corresponding cell states showed in horizontal bar plots. Gene expression has been normalized, log-transformed, and z-score transformed. Cell clusters are not predefined by developmental time-points. (E) Pathways enriched in immature and mature secretory cells revealed by gene ontology analysis. Selected top terms for biological process are shown. (F) Pathways enriched in immature and mature ciliated cells revealed by gene ontology analysis. Selected top terms for biological process are shown.

The online version of this article includes the following source data and figure supplement(s) for figure 3:

**Source data 1.** Gene Lists for scGPS analysis for cell cycle scoring.

**Figure supplement 1.** Conserved precursor states of ciliated cells.

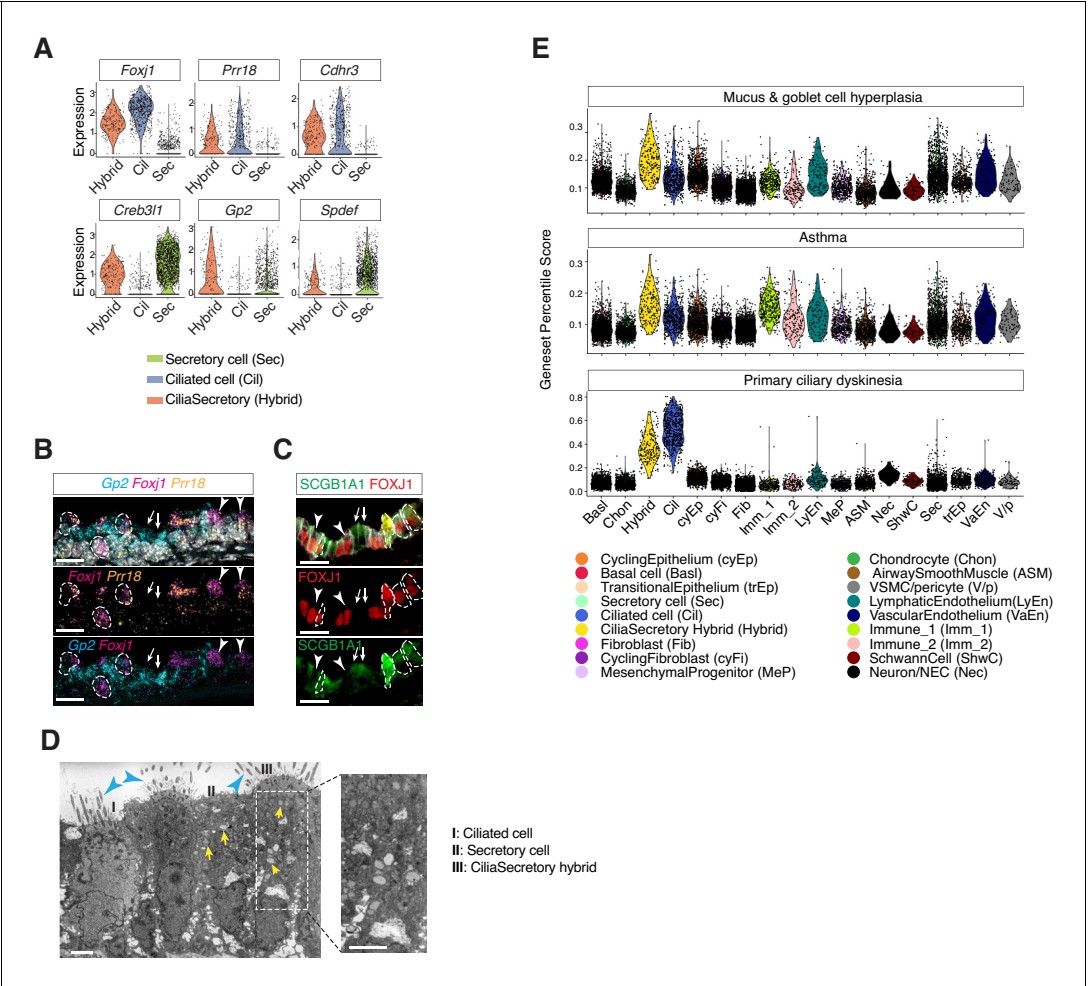

**Figure 4.** Characterization of neonatal cilia-secretory hybrid cells. (A) Violin plots showing the expression levels of ciliated cell markers *Foxj1*, *Prr18*, and *Cdhr3*, as well as secretory cell markers *Creb3l1*, *Spdef*, and *Gp2* in three cell clusters: ciliated cells, secretory cells, and cilia-secretory hybrids. Gene expression has been normalized and log-transformed. (B) RNA FISH analysis of *Foxj1* (magenta), *Gp2* (blue), and *Prr18* (yellow) mRNA in P3 wild-type trachea. Hybrid cells expressing all three markers are indicated by dashed circles. *Foxj1*$^+$ *Prr18*$^+$ ciliated cells are indicated by arrowheads. *Gp2*$^+$ secretory cells are indicated by arrows. Nuclei are marked by DAPI (grey). Scale bar indicates 20 μm. (C) Fluorescent immunostaining of FOXJ1 (red) and SCGB1A1 (red) in P3 wild-type trachea. Hybrid cells expressing both markers are indicated by dashed circles. FOXJ1$^+$ ciliated cells are indicated by arrowheads. SCGB1A1$^+$ secretory cells are indicated by arrows. Cell membranes are marked by E-cadherin (grey). Scale bar indicates 20 μm. (D) Transmission electron microscopy (TEM) images of P0 tracheal epithelial cells. Intracellular vesicles are indicated by arrows in yellow. Motile cilia are indicated by arrowheads in blue. Scale bar corresponds to 2 μm. (E) scGPS of disease-associated genes curated from the Online Mendelian Inheritance in Man (OMIM) database. Each dot represents a cell. Colors indicate cell clusters. Full lists of genes used are in *Figure 4—source data 1*. The online version of this article includes the following source data and figure supplement(s) for figure 4:

**Source data 1.** Gene Lists for scGPS analysis for airway diseases associated genes.
**Figure supplement 1.** Characterization of cilia-secretory hybrid cells.

cell types, and our dataset enables mapping of each disease mediator to its contributing cellular source.

Using scGPS (single-cell Geneset Percentile Score), an expression enrichment analysis that utilizes sets of genes underlying certain biological and pathological processes to infer functional profiles for each cell, we assessed whether the function of specific cell types can be inferred from the expression patterns of genes whose loss-of-function results in airway pathophysiology. Using genes curated by OMIM, we first assessed enrichment patterns for genes implicated in mucus and goblet cell hyperplasia (*Figure 4E*; *Figure 4—source data 1*), which are typically present in inflammation and infection and are classic symptoms for COPD and CF. Scores for mucosal hyperplasia are higher in cilia-secretory hybrid cluster, and are also elevated in T cells, endothelial cells, and vascular smooth

muscles (*Figure 4E*). Similar enrichment profiles are observed for asthma associated genes, indicating shared cellular drivers for goblet cell hyperplasia and asthma (*Figure 4E*). Primary ciliary dyskinesia (PCD) is a monogenic disorder associated with impaired motile cilia function. Patients of PCD show persistent rhinitis and chronic respiratory tract infections, but are often initially diagnosed as asthma or bronchiectasis (*Lucas et al., 2014*; *Shapiro et al., 2018*). The PCD scores are higher overall for ciliated cells and cilia-secretory hybrid cells, confirming that ciliated cells are primary targets for PCD. The results of our scGPS analysis provide functional validation of our cell type annotation, and help disambiguate cell-type involvement in the manifestation of monogenic and complex-trait diseases. Interestingly, the cilia-secretory hybrid cells show prominent enrichment scores in multiple airway conditions (*Figure 4E*), indicating a critical role of this hybrid cell state in airway function and pathogenesis.

## Immune profiles of the mucosal barrier in wild-type and *Ano1 KO* mutants

After the neonates transition from amniotic fluid to air breathing, their airway epithelial cells are in direct contact with the external environment and form the front line of innate host defense by producing a diverse arsenal of antimicrobial molecules and cytokines (*Ganesan et al., 2013*; *Vareille et al., 2011*). Our analysis revealed the specific cellular source and temporal expression patterns of a set of immune modulators important for mucosal barrier function and airway immunity (*Figure 5A*). Based on their functional profiles, these epithelial-derived molecules can be divided into two major groups. The first group consists of secreted peptides involved in the recognition of inhaled pathogens and mucin components (*Figure 5A*). The second group includes cytokines and signaling molecules implicated in mucosal inflammatory responses (*Figure 5A*). These molecules include *Cxcl15* (*Chen et al., 2001*), *Cxcl17* (*Burkhardt et al., 2014*), and *Ccl28* (*Hieshima et al., 2003*), which are chemokines involved in mucosal barrier function, *Nfkbia* and *Nfkbiz*, which are inhibitors for the NF-κb pathway serving as pivotal mediators of inflammatory responses (*Liu et al., 2017*; *Zhang et al., 2017*), as well as *Ptgs2*, *Ptges* and *Ptgds*, which are required for the biosynthesis of prostaglandin, a potent agent in the generation of the inflammatory response (*Ricciotti and Fitz-Gerald, 2011*).

In wild-type mice, we noticed an upregulation of *Retnla*, *Itln1*, and *Chil4* in the newborn airway epithelium (*Figure 5B*; *Figure 5—figure supplement 1A*). Both *Retnla* and *Chil4* are mediators of the Th2 inflammatory response, and they promote the formation of healthy barrier microbiota (*Harris et al., 2019*; *Nair et al., 2009*; *Nair et al., 2005*; *Pesce et al., 2009*). Significant downregulation of these markers was observed in all *Ano1*$^{-/-}$ mutant luminal cells (*Figure 5C*; *Figure 5—figure supplement 1A*). Conversely, mutant epithelial cells expressed high levels of *Cxcl17* and *Ccl20* (*Figure 5C*), two potent chemotactic factors for lung macrophages (*Burkhardt et al., 2014*; *Schutyser et al., 2003*). Mutant epithelial cells significantly downregulated NF-κB inhibitors *Nfkbia* and *Nkfbiz* (*Figure 5B,C*; *Figure 5—figure supplement 1A*), indicating that the NF-κB signaling may be abnormally regulated in the absence of *Ano1*. Consistent with the abnormal expression patterns of barrier defense genes, *Ano1*$^{-/-}$ mutant immune cells upregulated a set of immune modulators, including *Saa3*, that are expressed during inflammation (*Eklund et al., 2012*; *Figure 5D*). The expression levels of these barrier defense genes were comparable between wild-type and *Ano1*$^{-/-}$ mutant airway during developmental at E16 (*Figure 5—figure supplement 1B*).

## *Removal of Ano1* shifts basal progenitors toward differentiation into the secretory lineage

Secretory cells are one of the major cellular sources of mucin components and antimicrobial peptides in newborns. *Ano1*$^{-/-}$ mutants displayed an expansion of the secretory cell population across all time points we analyzed, from E16 to P1 and P4 (*Figure 6—figure supplement 1A*). Our data suggest an early requirement for *Ano1* expression during airway differentiation. *Ano1* mRNA can be detected as early as E12 in the respiratory epithelium (*Rock et al., 2008*). Using RNA FISH analysis and scRNAseq, we confirmed *Ano1* expression in undifferentiated mouse airway epithelium as well as in differentiated secretory cells (*Figure 6A,B*). Similarly, *CFTR* mRNA was broadly distributed in the mouse airway at E15 as well as in human fetal airway epithelium at GW15 (*Figure 6A,B,C*) and

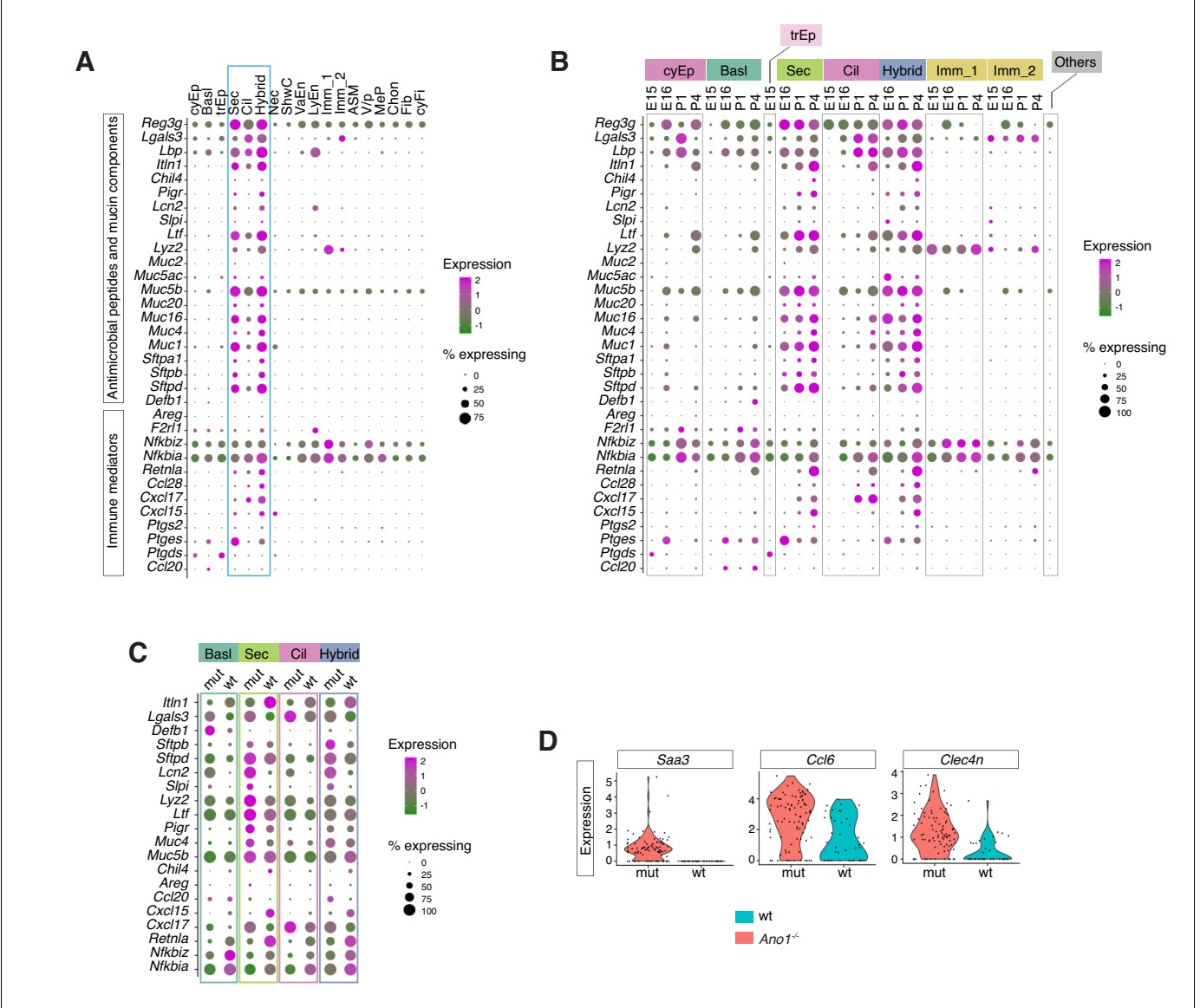

**Figure 5.** Ano1 KOs exhibit abnormal immune profiles of the airway mucosal barrier. (**A**) Dot plot showing the expression of mucosal barrier regulators and immune mediators. Legends for cell types are indicated. The size of the dot encodes the percentage of cells expressing the gene, while the color encodes the mean of expression level which has been normalized, log-transformed, and z-score transformed. (**B**) Dot plot showing the temporal expression of airway barrier genes. The size of the dot encodes the percentage of cells expressing the gene, while the color encodes the mean expression level after normalization, log-transformation, and z-score transformation. cyEp: Cyling epithelium, Basl: Basal cells, trEp: Transitonal epithelium, Sec: Secretory cells, Cil: Ciliated cells, Hybrid: Cilia-Secretory hybrid cells, Imm_1: *Fcer1g*[+]/*Cd3g*[+] immune cells, Imm_2: *Cx3cr1*[+]/*C1qa*[+] immune cells, Others: all other cell types and states (**C**) Expression of selected airway barrier genes altered in *Ano1*[-/-] mutant epithelial cells at P4. The size of the dot encodes the percentage of cells expressing the gene, while the color encodes the mean of expression level which has been normalized, log-transformed, and z-score transformed. Cell type legends for **A** and **B** are shown in the figure. (**D**) Violin plots showing the expression levels of *Saa3*, *Ccl6*, and *Clec4n* in the *Fcer1g*+ resident immune cells. Gene expression has been normalized and log-transformed.

The online version of this article includes the following figure supplement(s) for figure 5:

**Figure supplement 1.** Cell-type-specific expression of barrier function related genes in wild-type and *Ano1*[-/-] mutant tracheal epithelial cells.

expressed in secretory cells associated with the surface epithelium and submucosal glands at GW21-23 (*Figure 6C,D*).

To understand the mechanism of *Ano1*-dependent secretory cell differentiation, we analyzed cell states of different epithelial populations from E16. At this stage, secretory cells include a basal-to-secretory transition state and an immature secretory state, both of which are characterized by high

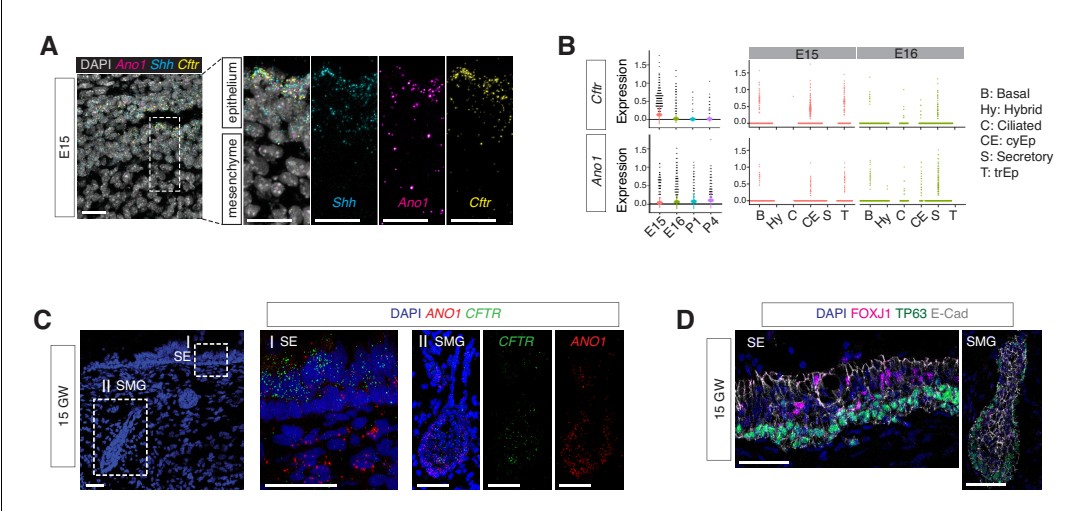

**Figure 6.** Expressions of *Ano1* and *Cftr* in embryonic and fetal airways. (**A**) Expression of *Shh*, *Cftr*, and *Ano1* in E15 trachea examined by FISH. *Shh* marks tracheal epithelial cells. Nuclei are stained by DAPI (white). Scale bar indicates 20 μm. (**B**) Expression of *Cftr* and *Ano1* in tracheal epithelial cells at different time points, with a cell type breakdown for E15 and E16. Expressions of *Cftr* and *Ano1* in neonatal tracheal epithelial cells are shown in *Figure 6—figure supplement 1*. (**C**) Expression of *CFTR* and *ANO1* in human fetal trachea of GW 15 by FISH. Two areas (I: surface epithelium/SE; II: submucosal glands/SMG) of the fetal tracheal epithelium are enlarged. Nuclei are stained by DAPI (blue). Scale bar indicates 50 μm. (**D**) Immunofluorescence staining of *TP63* (green) and *FOXJ1* (magenta) in human fetal tracheal sample at GW 15. SE and SMG are shown. Ciliated cells are only present in the surface epithelial cell layer. Epithelial cells are marked by E-cad in white. Nuclei are stained by DAPI (blue). Scale bar indicates 50 μm.

The online version of this article includes the following figure supplement(s) for figure 6:

**Figure supplement 1.** Expansion of the secretory cell lineage in global and conditional *Ano1* knockouts.

expression of *Krt4* (*Figure 7A*). Pseudotemporal analysis indicates that these two populations represent transitioning stages on the trajectory of basal cells differentiating into mature secretory cells (*Figure 7B*). The more committed secretory cells express high levels of mature secretory markers, such as *Muc5b*, *Creb3l1* and *Gp2*, as well as enzymes involved in mucin glycosylation, such as *Galnt6* and *B3gnt6* (*Figure 7A*). Compared to wild-type mice, the percentage of cells from the *Krt4*[+] immature secretory state markedly increased in *Ano1*[-/-] mutants (*Figure 7C*). We confirmed the expansion of these *Krt4*[+] immature secretory cells in neonatal mutant trachea via KRT13 immunofluorescent staining, given that *Krt13* is paired with *Krt4* in airway epithelial cells (*He et al., 2015*; *Montoro et al., 2018*; *Figure 6—figure supplement 1C*).

We identified a basal cell state marked by *Ccl20*, *Smoc2*, *Krt14*, as well as *Sostdc1*, a BMP and WNT pathway antagonist (*Kassai et al., 2005*; *Närhi et al., 2012*; *Figure 7A*). Because *Sostdc1*, *Smoc2*, and *Krt14* are expressed in various epithelial progenitor cells, we reasoned that this *Sostdc1*-expressing basal state represents a progenitor pool that gives rise to luminal cells during development and upon injury. In *Ano1*[-/-] mutants, the percentage of *Sostdc1*[+] basal cells was reduced (*Figure 7C*) and *Sostdc1* expression was significantly downregulated (*Figure 7D,E,F*). Changes associated with cell states were unlikely caused by altered cell proliferation, because cell cycle scores were comparable between wild-type and *Ano1*[-/-] mutants for basal and secretory cell states identified at E16 (*Figure 6—figure supplement 1B*).

To test whether the depletion of progenitor cells causes the expansion of the secretory population, and whether the inhibitory effect of *Ano1* on secretory cell differentiation is cell-autonomous, we generated two conditional *Ano1* KO mouse lines. First, we used *Shhcre*, which is expressed in all *Shh*-expressing endodermal cells during embryogenesis (*Swarr and Morrisey, 2015*), and found that embryonic removal of *Ano1* from the airway endoderm resulted in mucus cell hyperplasia and an expansion of *Krt13*[+] immature secretory cells (*Figure 7G*; *Figure 6—figure supplement 1D*). Low levels of *Ano1* mRNA expression are observed in basal progenitor cells in both mouse and human adult airway epithelial cells (*Figure 7—figure supplement 1*). Next, to determine whether *Ano1* plays a role in airway epithelium regeneration, we generated a basal cell-specific *Ano1*

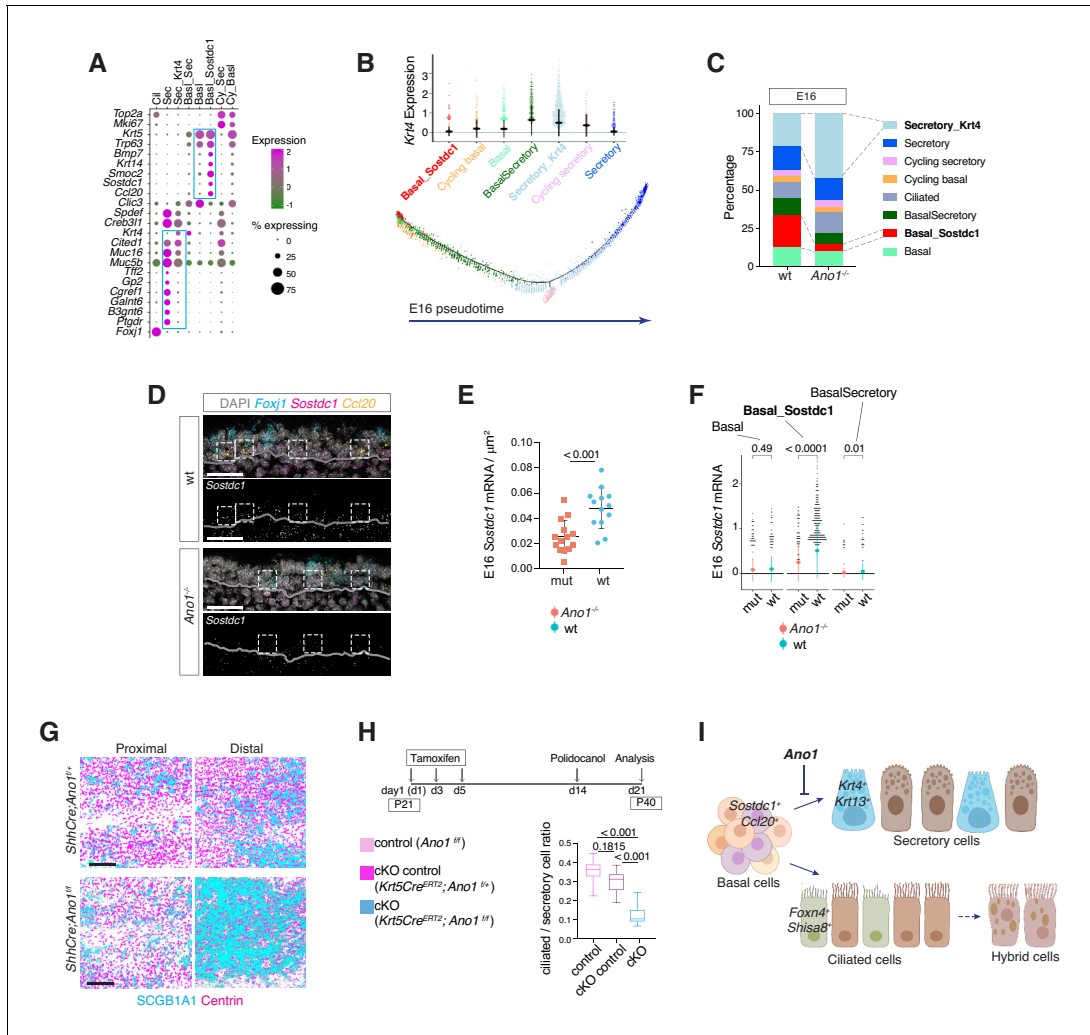

**Figure 7.** *Ano1* inhibits differentiation towards the secretory lineage in development and regeneration. (A) Dot plot depicting expression of marker genes of cell types and states identified from E16 tracheal epithelial cells. Markers for Basal_Sostdc1 and secretory cells are highlighted in blue box. The size of the dot encodes the percentage of cells expressing the gene, while the color encodes the mean of expression level that has been normalized, log-transformed, and z-score transformed. Both wild-type and *Ano1⁻/⁻* mutants are included for this analysis. Cil: ciliated cells; Sec: Secretory cells; Sec_Krt4: *Krt4⁺* immature secretory cells; Basl_Sec: basal-to-secretory transition state; Basl: basal cells; Basl_Sostdc1: *Sostdc1⁺* basal cells; Cy_Sec: cycling secretory cells; Cy_Basl: cycling basal cells. (B) Upper panel: *Krt4* expression in all basal and secretory cell states at E16. Expression values have been normalized and log-transformed. Lower panel: pseudotime trajectory of these cell states. Color codes for different cell states are indicated. (C) Cellular composition of tracheal epithelial cells from wild-type and *Ano1⁻/⁻* mutants at E16. (D) Expression of *Sostdc1*, *Ccl20*, and *Foxj1* in wild-type and *Ano1⁻/⁻* mutant trachea samples at E16 detected by FISH. *Sostdc1* single channel images are shown in black and white. Dashed squares indicate *Sostdc1⁺ Ccl20⁺* double-positive cells. Scale bar indicates 40 μm. (E) Concentration of *Sostdc1* RNA probes from wild-type and *Ano1⁻/⁻* mutant trachea samples at E16. n = 3 for each genotype. p-Value (unpaired two-tail t-test) are indicated. Error bars represent S.D. (F) Expression level of *Sostdc1* in wild-type and *Ano1⁻/⁻* mutants at E16 in different basal cell states. Each dot represents a cell. Expression values have been normalized and log-transformed. Colored circles indicate mean expression values. Colored vertical lines cover the range of one standard deviation above or below the mean. Adjusted p-values for the comparison within each cell type (unpaired two-tailed wilcoxon-test between wild-type and mutant) are indicated. (G) Distal tracheal epithelial cells exhibit mucous metaplasia in adult *ShhCre; Ano1^flox/-* and *ShhCre; Ano1^flox/flox* conditional knockouts at P25. Ciliated cells are marked by Centrin (magenta). Secretory cells are marked by SCGB1A1 (blue). n = 3 for each genotype. Scale bar indicates 50 μm. (H) Ratio of ciliated cells over secretory cells in regenerating adult trachea. Injection scheme of Tamoxifen and application of Polidocanol were indicated. *Ano1^flox/flox*, *Krt5Cre^ERT2; Ano1^flox/+* and *Krt5Cre^ERT2; Ano1^flox/flox* were included in this analysis. Both proximal and distal trachea were included in the analysis. n = 3 for each genotype. P-value (Ordinary one-way ANOVA Tukey test and multiple comparisons) are indicated. Box and whisker plot shows 10–90 percentile. (I) Model of *Ano1*-mediated epithelial cell differentiation of the embryonic trachea.

The online version of this article includes the following figure supplement(s) for figure 7:

**Figure supplement 1.** Expression of *Ano1* in postnatal airway epithelial cells.
**Figure supplement 2.** Single-cell velocity estimates for individual ciliated cells at E16.

conditional knockout mouse line by crossing $Ano1^{f/f}$ with inducible $Krt5Cre^{ERT2}$, which is activated upon intraperitoneal injection of tamoxifen and expressed in $Krt5^+$ airway progenitor cells (*Van Keymeulen et al., 2011*). In a polidocanol induced airway injury model (*Liu et al., 2006*), removal of *Ano1* from 3 week-old mice similarly resulted in a differentiation bias toward the secretory lineage during the regeneration phase (*Figure 7H*).

In contrast to the paradigm of airway cell fate specification, in which Notch signaling regulates the balance between ciliated cells and secretory cells, we did not observe a significant change in the percentage of ciliated cells in the absence of *Ano1*. The transcriptome profiles of ciliated cells are comparable between wild-type and $Ano1^{-/-}$ mutants (*Figure 7—figure supplement 2A*). Additionally, mRNA velocity analysis suggested that both genotypes exhibited the same developmental trajectory for ciliated cells at E16 (*Figure 7—figure supplement 2B*), in which $Foxn4^+$ and $Shisa8^+$ precursor states transition into more mature $Cdhr3^+$ and $Cfap53^+$ states (*Figure 7—figure supplement 2C,D*).

Overall, the percentage, transcriptome profiles, and developmental trajectory of ciliated cells are comparable between wild-type control and *Ano1* mutants. Based on the aforementioned changes in the cellular composition and molecular profiles between wild-type and *Ano1* mutant tracheal epithelial cells, we propose that *Ano1*-mediated chloride homeostasis maintains a *Sostdc1*-expressing progenitor pool within the epithelial basal cells, loss of which results in secretory hyperplasia at least partially due to an accumulation of intermediate secretory cells with elevated levels of *Krt4* and *Krt13* (*Figure 7I*). Lineage specification of ciliated cells, on the other hand, appears unaffected in the absence of *Ano1*.

## Discussion

Deviations from the normal developmental process can be detrimental to airway function in youth and may have long-lasting effects later in life. In this study, we present a high-quality single-cell atlas of the developing airway with a rich repertoire of cell types, including epithelial cells, a large collection of mesenchymal cells, immune cells, endothelial cells, and neuronal cells that can be further divided into more fine-grained and novel cell states associated with distinct developmental stages and functional profiles, reflecting the structural and physiological complexity of the conducting airway. Importantly, all cell types identified in the mouse trachea have corresponding orthologous cell types in humans, indicating that the two species employ conserved gene regulatory programs to build the large airway during embryogenesis.

Our assessment of enrichment patterns for selected airway disease genes using scGPS confirmed that most monogenic disease genes implicated in PCD are specifically expressed in ciliated cells. In contrast, susceptibility loci for COPD are expressed in both epithelial and stromal cell types, consistent with the pleiotropic presentations of complex-trait airway disease. The enrichment profiles generated by scGPS thus provide a framework for studying cell-type-specific contributions in the pathogenesis of complex respiratory diseases.

Spanning four developmental time points, our dataset uncovers milestones of trachea development from the onset of differentiation, through cell fate determination during embryogenesis, and into the air breathing transition at birth. Compared to the homeostatic and regeneration phases in adults, the developing trachea exhibits both quantitative and qualitative differences in several key aspects of its transcriptional landscape. First, embryonic tracheal tissue expresses high levels of cell cycle markers and shows a higher percentage of progenitor cells, including the double positive *Sostdc1-* and *Smoc2*-expressing basal progenitor cells located within the epithelium. Second, we identified gene modules for different cell states that may account for stage-dependent functional profiles. For example, a conserved precursor state for ciliated cells is marked by *Foxn4* and the novel marker *Shisa8*. *Foxn4* is required to promote motile cilia formation in *Xenopus* epidermis (*Campbell et al., 2016*), and its expression is very transient during frog embryogenesis (*Briggs et al., 2018*). Our data indicate that *Foxn4* expression peaks transiently at E16 in the mouse trachea as ciliated cells begin to emerge, supporting a central role of *Foxn4* in promoting motile ciliogenesis in mammals. Third, we uncovered two critical transcriptional events for the establishment of the mucosal barrier. During embryogenesis, epithelial cells upregulate multiciliated gene modules and secretory gene modules to initiate a massive differentiation process between E15 and E16. After the animals are born and start breathing, tracheal epithelial cells upregulate a set of mucosal

cytokines, antibacterial effectors, and Th2 immune response genes that are critical for the maturation of barrier function.

In the neonatal airway, we identified a cilia-secretory hybrid cell state that exhibits the combined molecular profiles of both ciliated and secretory cells. Our data indicate that this neonatal cell state is likely derived from a ciliated cell lineage. The hybrid state appears to be very rare during the adult homeostatic state and cannot be induced by chemically-triggered airway regeneration (*Rawlins et al., 2007*). In contrast, a cluster of cilia-secretory hybrid cells are found in the human inflammatory airway, indicating that neonatal airway may share similar immune status to the inflamed adult airway (*Ordovas-Montanes et al., 2018*; *Turner et al., 2011*; *Tyner et al., 2006*; *Vieira Braga et al., 2019*). In both neonatal mouse airway and asthma patients, hybrid cells are linked to the ciliated cell lineage. Supporting this notion, trans-differentiation of ciliated cells into goblet cells can be induced by IL-13, a major cytokine involved in allergy and inflammatory response (*Turner et al., 2011*; *Tyner et al., 2006*). While the in vivo functional attributes of this cell state have yet to be defined, the emergence of these hybrid cells may reflect a cellular plasticity in response to inflammation, which has been observed in the first moments of neonatal life (*Kollmann et al., 2017*; *Saluzzo et al., 2017*; *Torow et al., 2017*). This hybrid cell state may also play a role in fine tuning the balance between efficient mucociliary clearance and mucus production.

Using this atlas as a comprehensive and unbiased framework, and combining it with mouse genetic analysis, we identified the cellular origins of mucus cell hyperplasia and early onset barrier defects caused by loss of *Ano1*. First, *Ano1* in the undifferentiated epithelial cells plays an essential role in maintaining airway progenitors by limiting the differentiation of basal cells into the secretory lineage. Such activity of *Ano1* may be sufficient to account for the severe mucus cell hyperplasia observed in *Ano1*$^{-/-}$ mutants. Second, persistent expression of *Ano1* in differentiated airway epithelial cells controls the normal status of the neonatal mucosal immunity. A shift in the expression of antimicrobial genes and proinflammatory modules in *Ano1* mutants indicates that these mutants may be more prone to infection and inflammation, which result in secondary pulmonary dysplasia (*Davidson and Berkelhamer, 2017*; *Shahzad et al., 2016*). Intracellular chloride, regulated by chloride channels, has been implicated in vesicle trafficking and plays a role in regulating plasma membrane dynamics (*Bradbury et al., 1992*; *He et al., 2017*; *Stauber and Jentsch, 2013*). While the specific actions of *Ano1* in basal cell differentiation and barrier functions require further investigation, it is possible that *Ano1* mediated-chloride homeostasis underlie both aspects of *Ano1* in airway development and physiology.

Inactivation of *CFTR* in humans and *Ano1* in mice, respectively, lead to congenital abnormalities of the airway, raising the possibility that perhaps intracellular chloride homeostasis, modulated by these two chloride channels, represents a conserved component for mammalian airway development. Because the pathogenesis and clinical presentations of CF in humans are extraordinarily complex, and the mucosal immunities vary among species, animal models for airway disease can only recapitulate limited aspects of human symptoms. Despite these challenges, our study provides a conceptual foundation for the ontology of mammalian trachea, and demonstrates the presence of conserved cell types and gene modules present in the mouse and human conducting airway. Notwithstanding the challenge in reconciling the discrepancies in mouse and human CF pathogenesis, our tractable mouse model allows for discoveries of airway cell types that require chloride channels for proper differentiation and functions that are relevant to early onset airway diseases.

## Materials and methods

### Mice

The *Ano1* null allele, *Ano1*$^{tm1Jrr}$, and *Tmem16a* conditional allele, *Ano1*$^{tm12Jrr}$, have been described previously (*Rock et al., 2008*). Breeding colonies were maintained in a mixed genetic background by outcrossing C57BL/6J *Ano1*$^{tm1Jrr}$ and C57BL/6J *Ano1*$^{tm1Jrr}$ males to FVB females, which were obtained from JAX. For lineage tracing, *Shh*$^{tm1(EGFP/cre)Cjt/J}$ (*Harfe et al., 2004*) and *Gt(ROSA) 26Sor*$^{tm4(ACTB-tdTomato-EGFP)Luo/J}$ (*Muzumdar et al., 2007*), *Krt5*$^{tm1.1(cre/ERT2)Blh}$ (*Van Keymeulen et al., 2011*), and *Foxj1*$^{tm1.1(cre/ERT2/GFP)Htg/J}$ (*Muthusamy et al., 2014*) were obtained from JAX. Arl13b-mCherry/Centrin-GFP reporter mice *Tg(CAG-Arl13b/mCherry)*$^{1Kv}$ and *Tg(CAG-EGFP/ CETN2)*$^{3-4Jgg/KvandJ}$ (*Bangs et al., 2015*) were obtained from Sloan Kettering Institute. Mice were

housed in an animal facility and maintained in a temperature-controlled and light-controlled environment with an alternating 12 hr light/dark cycle. A maximum of five mice were housed per cage. All protocols have been approved by the University of California San Francisco Institutional Animal Care and Use Committee.

### Isolation of mouse trachea cells

To obtain embryonic tracheal cells for scRNA-seq, pregnant female mice were sacrificed via $CO_2$ asphyxia at desired stages and embryos were collected. Neonatal mice were sacrificed by decapitation. Tracheas were collected and washed with ice cold DMEM/F12 1:1 (Gibco, 25200056) to remove residual blood. For single-cell dissociation of embryonic trachea, samples were dissociated with 0.25% Trypsin-EGTA and 0.1 mg/mL DNaseI in DMEM/F12 at 37°C for 15 min. For neonatal trachea, samples were incubated with a combination of 1 mg/mL elastase (Worthington, LS006363) and 2.5 mg/mL dispase II (Roche, 4942078001) in DMEM/F12 0.1 mg/mL DNaseI for 15 min, and then 0.125% Trypsin-EGTA for 15 min at 37°C. Digest reaction was terminated by an addition of 10% bovine calf serum. Dissociated single-cell solution was centrifuged at 300 g for 5 min at 4°C. Cell pellets were resuspended with cold DMEM/F12 with 5% bovine calf serum and passed through a 35 μm filter into a collection tube. Single cells were counted with a Neubauer chamber and cell viability was assessed with trypan blue staining. Using this protocol, we consistently obtained >90% viable cells. Cells expressing GFP and RFP dissociated from *mT/mG* samples were separated collected via Fluorescence Activated Cell Sorting (FACS) using an SH800S (Sony) sorters. Post-sorting, the cells were collected in cold DMEM/F12 media and immediately proceeded to library preparation.

### Isolation of human fetal tracheal cells

Human fetal trachea samples were obtained and used in accordance with the guidelines for the care and use of animals and human subjects at University of California, San Francisco. Given that deidentified fetal tissue were involved, this study does not involve human subjects as defined by the federal regulations summarized in 45 CFR 46.102(f) and does not require IRB oversight. Details for approval is included in the accompanying GDS certification letter from the UCSF Ethics and Compliance and the Human Research Protection Program with study ID number 16–19909. First and early second trimester human fetal trachea were collected without patient identifiers after elective termination of pregnancy with approval from the Committee on Human Research at UCSF (IRB#12–08813). Fetal age was estimated using heel-toe length (*Drey et al., 2005*). Fetal age was calculated from time of fertilization, fetal age, and not from last menstrual period. Fetal trachea samples were collected at room temperature during surgery and analysis (1–2 hr) before stored at 4°C. For dissociation, tissue pieces were rinsed in ice cold PBS and incubated in a combination of 200 μg/ml Liberase (Liberase TL Research Grade, Roche, 05401020001) and 0.1 mg/mL DNaseI in DMEM/F12 for 30 min. Digest reaction was terminated by addition of 10% bovine calf serum, followed by an additional step to remove red blood cells via RBC Lysis buffer (ThermoFisher). Following steps were identical to the one used for mouse single cell collection.

### Single-cell RNA sequencing

Single cells were encapsulated into emulsion droplets using the Chromium Controller (10x Genomics). scRNA-seq libraries were constructed using Chromium Single Cell 3' reagent kits v2 (mouse samples) or v3 (human samples) according to the manufacturer's protocol. About 3000 to 7000 cells were targeted in each channel. Reverse transcription and library preparation were performed on a C1000 Touch Thermal cycler with 96-Deep Well Reaction Module (Bio-Rad). Amplified cDNA and final libraries were evaluated on Agilent tapestation system (Agilent Technologies). Libraries were sequenced with 26 (v2) or 28 (v3) bases for Read1, eight bases for Index1, and 98 bases (v2) or 91 bases (v3) for Read2 on the Novaseq 6000 Sequencing System (Illumina) to over 80% saturation level.

### Annotations of cell types and cell states in the mouse airway

To define cell types, we analyzed all cells sampled from trachea of wild-type mice across ages by performing principal component analysis on the most variable genes between cells, followed by nearest-neighbor graph-based clustering (code available through https://github.com/czbiohub/

BingWu_DarmanisGroup_TracheaDevTmem16a; *Wu, 2020*; copy archived at https://github.com/eli-fesciences-publications/BingWu_DarmanisGroup_TracheaDevTmem16a). We then generated a list of gene expression markers for each cluster using a Wilcoxon Rank Sum test Based on gene expression of known markers, we then assign cell-type annotations to each cluster. To define the developmental dynamics of epithelial cell types, each epithelial cell type that is composed of multiple clusters, or cell states, are displayed in separate heatmaps. Within each heatmap, the cells are grouped by their original cluster identity and annotated with their cell states based on the differentially expressed genes without using developmental stage information. Genes shown in the heat maps are selected sets of differentially expressed genes of each cluster compared to the rest of the cells in the same cell type. The same lists of differentially expressed genes were uploaded to the STRING database search portal (https://string-db.org/) to perform gene ontology analysis.

## Antibodies and immunostaining

Antibodies for immunofluorescence staining were mouse anti-FOXJ1 (1:500, 2A5, ThermoFisher, 14-9965-82), rabbit anti-TRP63/P63 (1:500, proteintech, 12143–1-AP), SCGB1A1/CC10 (1:200, B6, Santa Cruz, sc-390313), rabbit anti-KRT13 (1:500, proteintech, 10164–2-AP), mouse anti-acetylated α-tubulin (1:2,000; 6–11B-1; Sigma-Aldrich T6793), rat anti-E-cadherin (1:1,000; ECCD-2; Thermo Fisher Scientific), rabbit anti-DCLK1 (1:500, ThermoFisher, PA5-20908), anti-GFP (1:1000, Aves lab, AB_2307313), Alexa Fluor 488-, 594- and 633-conjugated secondary antibodies (Invitrogen), and Fluorescein labeled Jacalin (1:500, Vector Laboratories, FL-1151). For protein immunostaining, cells or tissue sections were fixed with 4% paraformaldehyde (PFA) for 20 min at room temperature or −20C° methanol for 10 min on ice. After fixation, the samples were washed and blocked with IF buffer (1 × PBS with 1% heat-inactivated goat/donkey serum and 0.3% Triton X-100). Primary antibodies were added and incubated for 1 hr at room temperature or overnight at 4°C. After washing with IF buffer, secondary antibodies and DAPI were added at 1:1000 dilution for 1 hr at room temperature. Samples were washed with 1 × PBS and mounted with Fluoromount-G (SouthernBiotech). Washing and staining were performed with IF buffer (1 × PBS with 5% serum and 0.2% triton) at room temperature. Samples were then imaged using a Leica TCS SP8 confocal microscope with the 40 × and 63 × HC PL Apo oil CS2 objective.

## Transmission electron microscopy

For transmission electron microscope, embryonic and newborn tracheas were dissected in cold PBS and fixed with 2% paraformaldehyde and 2.5% glutaraldehyde in 0.1 M sodium cacodylate buffer. After buffer rinses, samples were postfixed in 1% $OsO_4$ at room temperature for 4 hr followed by dehydrating in an ethanol series. Samples were stained with osmium tetroxide and embedded for thin sectioning in EPON. Sections of 70–100 nm were examined on a JEOL transmission electron microscope and photographed at primary magnifications of 4000–30,000X.

## *Foxj1Cre^ERT2* lineage tracing

Tamoxifen (5 mg/ml; 100 µl per animal) via intraperitoneal injection was administered one time on E14.5 in *Rosa26mT/mG* females crossed with *Foxj1Cre^ERT2:GFP* carrier males. Trachea from *Foxj1-Cre^ERT2:GFP* carrier newborns were collected at P0 to P1. Before fixation, we isolated trachea samples under a fluorescent dissection scope and only processed those with both Tomato and GFP signals. Samples were fixed with 4% paraformaldehyde (PFA) for 20 min at room temperature followed by −20C° methanol for 10 min on ice. After fixation and whole-mount staining, samples were flat mounted on 35 mm No 1.5 MatTek dishes (luminal side facing the coverlslip) and imaged *en-face*. For staining, mGFP signal was used to indicate *Foxj1⁺* lineage. mTomato signal was quenched after a sequential fixation by PFA and methanol, so we used E-cad antibody to label cell membrane in the RFP channel and pseudo-colored it in grey in the image panel. A mouse monoclonal antibody for SCGB1A1 (Santa Cruz, sc-390313) was used to label secretory cells in the far-red channel. Whole-mount airway samples were imaged with a Leica TCS SP8 confocal microscope with the 40 × HC PL Apo oil CS2 objective.

## Polidocanol-induced airway regeneration

Tamoxifen (20 mg/ml; 500 ul per animal) via intraperitoneal injection was administered three times on days 1, 3 and 5 in *Ano1^{f/f}; mT/mG, Ano1^{f/+}; Krt5Cre^{ERT2};mT/mG*, and *Ano1^{f/f}; Krt5Cre^{ERT2};mT/mG* mice at P25. A week after the last injection, mice were anaesthetized and one dose of 10 μl 2% polidocanol was administered via a pipet to the larynx to induce airway luminal cell injury. Non-surgical intratracheal instillation was modified based on previous protocol (*Rayamajhi et al., 2011*). Trachea samples were collected a week following injury for fixation and immunofluorescence.

## Histology

Newborn trachea and lung were fixed by formalin, dehydrated, and embedded in paraffin. Standard Periodic acid–Schiff (PAS) staining for airway mucus was formed at Mouse Pathology Core at the University of California, San Francisco.

## Cilia flow analysis

For imaging of ciliary flows, tracheas from neonatal animals were dissected in ice-cold Dulbecco's Modified Eagle Medium: Nutrient Mixture F-12 (ThermoFisher, DMEM/F12), and sliced open along the proximal-distal axis prior to imaging. Each trachea was mounted onto 35 mm cell imaging dishes, luminal side facing the coverglass, (MatTek) with a solution of fluorescent beads (Carboxylate-modified Microspheres 0.4 μm size; Invitrogen, 1:250 to 1:500) in 500 ml DMEM/F12 media. Tissue samples with 500 μl imaging media were placed on the microscope for 2 to 3 min to equilibrate to 37°C prior to imaging. Imaging was performed in an environmental chamber with 5% $CO_2$ at 37°C and acquired using a Leica TCS SP8 confocal microscope with a10x HCX PL Apo dry CS objective. Flow was imaged at 1 f/s for 300 s. Flow pathlines were generated using Flowtrace (*Gilpin et al., 2017*) (http://www.wgilpin.com/flowtrace_docs/). Particle Image Velocimetry (PIV) fields were generated using PIVLab (https://www.mathworks.com/matlabcentral/fileexchange/27659-pivlab-particle-image-velocimetry-piv-tool) for MATLAB. We used the FFT window deformation PIV algorithm with three passes consisting of $128 \times 128$, $64 \times 64$, $32 \times 32$ interrogation areas to determine the velocity vectors. Velocity vectors were filtered around a Gaussian Distribution (within 0.8 standard deviation) around no movement to remove incorrect pairings and extraneous movements. All parameters were held constant across all analyses to reduce systematic errors due to inconsistent discretization.

## mRNAs fluorescent in situ hybridization (FISH)

In situ hybridization for mouse *Ano1*, *Shh*, *Cftr*, *Foxj*1, *Foxn4*, *Gp2*, *Prr*18, *Shisa8*, *Mcidas1*, *Sostdc1*, *Ccl20*, and human *ANO1*, *CFTR*, *FOXJ1*, and *FOXN4* were performed using the RNAscope kit (Advanced Cell Diagnostics) according to the manufacturer's instructions.

## Statistical analysis

Methods for statistical analysis and numbers of samples measured in this study are specified in the figure legends. The error bars indicate the SD.

## Data analysis

Sequences generated by the NovaSeq were de-multiplexed and aligned to the mm10.1.2.0 genome using CellRanger (10x Genomics) with default parameters. Subsequent filtering, variable gene selection, reduction of dimensionality, clustering, and differential expression analysis with Wilcoxon rank sum tests were performed using the Seurat package (version 2.3) in R.

## scGPS (single-cell Geneset percentile scoring)

The geneset percentile score for each cell was calculated as the mean of cell-wise percent rank for all genes in a certain module. For instance, a given module consists of the number of m genes, an scGPS= $\frac{1}{m}\sum_{i=1}^{m} PercentRank_i$, in which $PercentRank_i$ is the rank of the log-normalized expression level of $gene_i$ in this cell compared to $gene_i$ expression in all cells in the dataset. Equal values of $gene_i$ expression are assigned the lowest rank. Each ranking is scaled to [0,1]. An scGPS of p can be interpreted as the mean expression of the selected genes is p percentile for the given cell. Implementation in R

can be found in https://github.com/czbiohub/BingWu_DarmanisGroup_TracheaDevTmem16a (*Wu, 2020*).

## Velocyto analysis of ciliated cell dynamics

RNA velocity was estimated by following velocyto.py documentation. Spliced and unspliced transcript counts were derived from Cellranger's outputs and with 'run10x' default settings through velocyto.py command-line interface. Cell type annotations were determined as described in *Figure 7* with Seurat in R. Scripts to reproduce the results for E16 epithelial cell dynamics including the transition trajectory of ciliated cells are available at https://github.com/czbiohub/BingWu_DarmanisGroup_TracheaDevTmem16a (*Wu, 2020*).

## Acknowledgements

We thank James Webber from Chan Zuckerberg Biohub for support with RNA-seq data processing; Laurence Baskin of UCSF for providing access to human fetal samples; Kathryn Anderson of Sloan Kettering Institute for sharing Arl13bmCerry/Centrin-GFP transgenic mice; the Bios Core Histology and Biomarker Teams at UCSF for histology analysis; David Erle for insightful discussion and comments; Mario Zubia, Tong Cheng, Guillermina Ramirez-san Juan, and Jorge Alexis Vargas for reagents and tissue analysis. This work is supported by Chan Zuckerberg Biohub and the National Institutes of Health (R01NS069229 to LYJ and F32HD089639 to MH). LYJ and YNJ are investigators of the Howard Hughes Medical Institute.

## Additional information

### Funding

| Funder | Grant reference number | Author |
| --- | --- | --- |
| National Institute of Neurological Disorders and Stroke | RO1 NS069229 | Lily Yeh Jan |
| Eunice Kennedy Shriver National Institute of Child Health and Human Development | F32HD089639 | Mu He |
| Howard Hughes Medical Institute | | Yuh Nung Jan<br>Lily Yeh Jan |

The funders had no role in study design, data collection and interpretation, or the decision to submit the work for publication.

### Author contributions

Mu He, Conceptualization, Data curation, Formal analysis, Supervision, Funding acquisition, Validation, Investigation, Visualization, Methodology, Writing - original draft, Writing - review and editing; Bing Wu, Conceptualization, Data curation, Software, Formal analysis, Validation, Investigation, Visualization, Methodology, Writing - review and editing; Wenlei Ye, Data curation, Formal analysis; Daniel D Le, Methodology; Adriane W Sinclair, Rene Sit, Michelle Tan, Norma Neff, Resources; Valeria Padovano, Michael J Caplan, Resources, Investigation, Methodology; Yuzhang Chen, Ke-Xin Li, Formal analysis, Investigation; Yuh Nung Jan, Funding acquisition; Spyros Darmanis, Resources, Data curation, Supervision, Writing - review and editing; Lily Yeh Jan, Resources, Supervision, Funding acquisition, Writing - review and editing

### Author ORCIDs

Mu He https://orcid.org/0000-0002-3812-6165
Wenlei Ye http://orcid.org/0000-0002-4694-1493
Valeria Padovano http://orcid.org/0000-0002-0142-3385
Ke-Xin Li http://orcid.org/0000-0003-3879-294X
Yuh Nung Jan http://orcid.org/0000-0003-1367-6299
Lily Yeh Jan https://orcid.org/0000-0003-3938-8498

**Decision letter and Author response**
Decision letter https://doi.org/10.7554/eLife.53085.sa1
Author response https://doi.org/10.7554/eLife.53085.sa2

## Additional files

### Supplementary files
• Transparent reporting form

### Data availability

Sequencing reads and processed data in the format of gene-cell count tables are available from the Sequence Read Archive (SRA) (SRA accession: PRJNA548516). All codes used for analysis in this study are available on GitHub (https://github.com/czbiohub/BingWu_DarmanisGroup_TracheaDevT-mem16a; copy archived at https://github.com/elifesciences-publications/BingWu_DarmanisGroup_TracheaDevTmem16a).

The following dataset was generated:

| Author(s) | Year | Dataset title | Dataset URL | Database and Identifier |
|---|---|---|---|---|
| He M, Wu B, Ye W, Le DD, Sinclair AW, Padovano V, Chen Y, Li KX, Sit R, Tan M, Caplan MJ, Norma Neff, Jan YN, Darmanis S, Jan LY | 2020 | trachea development | http://www.ncbi.nlm.nih.gov/bioproject/?term=PRJNA548516 | NCBI BioProject, PRJNA548516 |

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
