## [Decision Letter]

Thank you for sending your article entitled "Chloride channels regulate differentiation and barrier functions of the mammalian airway" for peer review at *eLife*. Your article is being evaluated by Edward Morrisey as the Senior Editor, a Reviewing Editor, and three reviewers.

Given the list of essential revisions, including new experiments, the editors and reviewers invite you to respond within the next two weeks with an action plan and timetable for the completion of the additional work. We plan to share your responses with the reviewers and then issue a binding recommendation.

We ask that you address both the technical limitations raised by the reviewers (i.e. lack of methodological details, additional details on the morphological defects in *Tmem16* null lungs etc.), and how you will provide additional data on the relationships between your described hybrid cells, ciliated, and secretory cells of the airways.

*Reviewer #1:*

In this paper the authors investigate the role of the chloride channel *Ano1/Tmem16a* in airway development and provide a survey of the cell types in the developing mouse and human tracheas using a scRNAseq and computational approaches. They report broad effects of *Tmem16a* deficiency, ranging from mucus hyperplasia to alveolar simplification. Their survey describes not previously reported hybrid cilia-secretory cells as well as novel immune and mesenchymal cells. They conclude that *Tmem16a* has non-redundant functions in the airway epithelium and that human and mice have conserved gene regulatory programs of airway differentiation.

The paper has an impressive amount of data and use of resources. However, the analysis of the data and integration into a coherent picture is overall quite superficial. Much of the data presented and the methodologies are barely described and the information is diluted, making it difficult to conclude about how the chloride channel orchestrate the changes observed or any potential link with CF.

1) The authors report in Figure 1A, B that *Tmem16a^-/-^* knockout results in mucus obstruction of the respiratory tract and alveolar simplification in neonatal mutants. What is provided is poorly documented and insufficient. The PAS staining is of poorly quality to demonstrate mucus obstruction; the goblet cell metaplasia needs to be further documented by Alcian Blue and Muc5a stainings. The region depicted by the square (I 2B WT), looks rather a blood vessel and is very different from the airway shown I 2B in the mutant lung. Moreover, from the picture provided in Figure 1B there is no convincing evidence of alveolar simplification in mutant, which should be shown by morphometrical analysis. It is unclear whether the authors mean to report a defect in sacculation at P1. The alveolar phenotype should be confirmed at P4 or later, since alveolar formation initiates at around P3. How do the authors explain this phenotype? Is Tmem16a expressed in the alveolar compartment?

2) What is the significance of the reversed mucociliary flow (0.42 ± 0.18 μm/s) in *Tmem16a* mutants and how could this be mechanistically explained and associated with the phenotype?

3) There is insufficient information about how the parameters depicted in Figure 2 were analyzed and displayed. For example, in Figure 2B the heatmap shows average expression of marker genes for each cluster identified at E15, E16, P1 and P4. However, there is no indication of how gene expression at the ages indicated are displayed. Methods and result sections are vague and uninformative. Same for the human samples in Figure 2C.

4) There is no lineage study or other evidence to support the authors' statement that the hybrid ciliated-secretory cells derive from ciliated cells. There are no definitive follow up experiments to determine whether these cells represent a transient state and whether they originate from secretory or ciliated cells. As described, the analysis of *Pofut1^-/-^* in Figure 4—figure supplement 1 neither convincingly demonstrate the existence of these hybrids cells, nor identify their origin. The Arl13b-mCheery/Centrin-GFP reporter mice are mentioned in the Materials and methods section but data are not shown or discussed.

5) *Spdef* and *Creb3l1* are not involved in the specification of embryonic (E16) secretory cells. In fact, *Spdef* expression is reported to initiate later in the developing lung and postnatally. Moreover, *Spdef* is not associated with induction of secretory (Club). In fact, *Spdef* inhibits Club cell differentiation and induces the goblet cell program (Park et al., 2007).

6) It is not clear what the authors mean by "By demonstrating the similarities in airway defects as well as in expression patterns of *Tmem16a* and *CFTR* in mouse and human airway epithelium, respectively, our work provides an entry point for understanding the developmental aspects of airway barrier pathogenesis". In spite of the focus on chloride channels and comparisons with *CFTR*, at the end it unclear what *Tmem16a* is really doing in these cells.

*Reviewer #2:*

The manuscript "Chloride channels regulate differentiation and barrier functions of the mammalian airway" by Mu He et al. reports the role of *Tmem16a/Ano1* in maintaining airway barrier function. The deletion of *Tmem16a* in mouse was shown to result in mucus obstruction and defective mucociliary clearance during development. Through transcriptional analyses, the authors characterized the developmental landscape of airways in both mouse and human and showed that *Tmem16a* has a role in regulating inflammation and epithelial progenitor cell differentiation. Collectively, the authors conclude that chloride channels are essential for mammalian airway formation and function.

While roles for *Tmem16a* in lung trachea development have been reported previously, a role for *Tmem16a* in airway epithelial development has not been reported. In addition, the authors identified remarkably conserved cellular programs operating during human fetal lung development. Therefore, the observations in the manuscript have novelty and are interesting. There are however several comments that I have listed below that I feel can improve the manuscript.

1) Figure 1A showed that *Tmem16a* null mutant mice had increased mucus secretion. Are the levels of submucosal glands and goblet cells increased in the mutant?

2) Figure 1E showed increased number of SCGB1A1^+^ cells in *Tmem16a* null mutants. It is unclear whether *Tmem16a* impacts cell proliferation. Does *Tmem16a* deletion increase SCGB1A1^+^ cell proliferation?

3) Figure 6A, B showed *Tmem16a* expression in E15 mouse lung epithelium. It is unclear whether *Tmem16a* is expressed in postnatal lung epithelium during both homeostasis and Polidocanol-induced airway regeneration.

4) Figure 4C showed that *Tmem16a* null mutant epithelial cells significantly downregulated NF-κB inhibitors *Nfkbia* and *Nkfbiz*. It is unclear whether decreased *Nfkbia* and *Nkfbiz* globally impacts NF-κB signaling. Does *Tmem16a* deletion increase NF-κB nuclear activity in lung epithelial cells?

5) *Tmem16a* null mutants displayed an expansion of immature secretory cell populations. I wonder if this phenotype was caused directly by the loss of *Tmem16a* or indirectly via increased inflammation. Does reduced inflammation recue the phenotype observe in the *Tmem16a* null mutant?

6) Figure 6H showed that deletion of *Tmem16a* in adult *Krt5*^+^ cells resulted in decreased ratio of ciliated cells versus secretory cells during Polidocanol-induced airway regeneration. Do the Krt5-creERT2, *Tmem16a* fl/fl mice exhibit similar phenotypes as those observed in *Tmem16a* null mutants, such as mucus obstruction and increased inflammation?

7) Figure 6G showed that deletion of *Tmem16a* in mouse epithelium led to mucous metaplasia and biased differentiation toward secretory cells during postnatal life. It's interesting that prior studies have indicated that *Tmem16a* overexpression led to mucous hyperplasia and blocking *Tmem16a* suppressed mucin secretion in the lung secretory epithelial cell (Huang et al., 2012). These studies suggest that both loss- and gain-of-expression of *Tmem16a* in lung epithelial cells could cause mucous hyperplasia. It is unclear about the reasons underlying these contradictory findings. The rationale and discussion are needed to clarify these findings.

*Reviewer #3:*

In this manuscript, He et al., studied the role of chloride channel *Ano1/Tmem16a* in airway development and its role in barrier function. The authors used *Tmem16* knockout mouse model and claim that inactivation of *Tmem16* results in alveolar simplification and obstruction of the respiratory tract due to mucus accumulation in neonatal mutants. To characterize the cellular origins of airway defects, the authors performed single-cell RNA sequencing of mouse trachea on wild-type and *Tmem16a* mutant cells from embryonic tissues. Further the authors profiled human fetal tracheal cells at gestation weeks 21 and 23 (GW21 and GW23) to identify orthologous cell types and cell states defined by similar molecular markers between the mouse airway and the human airway. Based on this data, the authors claim to have identified a novel cilia-secretory hybrid sell state using *Foxj1, Gp2* and *Prr18* as markers. Additionally, they state that these hybrid cells are likely derived from a ciliated cell lineage and play a critical role in airway function and pathogenesis including Asthma, COPD, and PCD. Finally, using scRNAseq the authors claim that loss *Tmem16a* results in secretory cell hyperplasia and *Tmem16a* inhibits basal progenitor differentiation towards the secretory lineage.

Overall, the authors used scRNA-sequencing to understand the development of airway epithelium and to study the role of *Tmem16* in this process. However, the data presented here is incoherent and lacks clear message. The data provided in the current manuscript is of poor quality and does not add much to the knowledge in this area. It appears to me that the authors put together two unrelated pieces (scRNA-seq and *Tmem16*). The authors point to a hybrid cell state that's representative of both ciliated and secretory cells. As pointed in the later section, the numbers and proportions of these intermediates is unclear, and the validation data presented here is not convincing. Similarly, the authors talked about genes enriched in CF, COPD, asthma and COPD but there is no direct correlation to any of the phenotypes described here.

Essential revisions:

1) *Tmem16* mutant characterization: In Figure 1, the authors present data from *Tmem16* mutant airway tissue characterization. The authors compared the muco-ciliary clearance between the wild-type and *Tmem16a^-/-^* knockout mouse line without describing any experimental details. In Figure 1A, the authors used Jacalin-488 staining. However, it is unclear why the authors chose to use Jacalin as this is not a commonly used marker for airway cell types. From the images, it appears that it marks all luminal cells of the airway epithelium. Co-staining with secretory and ciliated cell markers would help. In Figure 1B, PAS staining does not look convincing – a better representative image can be included. Further, authors talked about defects in mucociliary clearance (Figure 2C) by assessing the flow dynamics of fluorescent beads in *Tmem16a^-/-^* mouse without describing the tracheal abnormalities in *Tmem16a^-/-^* knockout mouse? What are the anatomical and structural differences between wild-type trachea to *Tmem16a^-/-^* knockout? Previous studies have shown that the tracheal cartilage rings are abnormal in the mutants. So, is it possible that the MCT defects observed here are due to abnormalities in tissue structure? And also, some of the phenotypes shown here have been already described elsewhere. The authors failed to cite the previous studies in this section.

2) Correlation between *Tmem16* expression and function: In the current study authors used embryonic and neonatal mouse tissue and studied the role of *Tmem16a* in formation and function of the airway mucosal barrier without clearly characterizing the *Tmem16a* expression pattern at different developmental stages. Authors state that *Tmem16a* is expressed in undifferentiated mouse epithelium as well as in differentiated secretory cells at E15 (Figure 6A-B). However, In the Figure 2E authors shows their single cell RNA-seq identify *Ano1/Tmem16a* is specific to the mouse secretory cells. Does that mean later in development or in neonates only secretory cells express *Ano1/Tmem16a*? if so, at which developmental stage *Ano1/Tmem16a* expression is specified to the secretory cells?

3) scRNA-seq data and hybrid cell state: The authors performed single-cell RNA sequencing of mouse trachea from wild-type embryonic day 15 (E15), E16, P1, and P4 and *Tmem16a* mutant trachea of E16, P1, and P4 to characterize the cellular origins of these complex airway defects. However, they did not show as to how does the comparative data look like? What are the major transcriptomic differences between the wild-type and *Tmem16* mutant trachea? How does the cellular landscape look like on *tSNE*/UMAP? Further, they assign the percent distribution of each cell population based on scRNA-seq data (Figure 2—figure supplement 1B). Does that represent the actual distribution of these cell populations in mouse trachea? Did authors account for any artifacts associated with tissue dissociation and sampling? Immunofluorescence data from Figure 3F and G indicates that the hybrid population accounts for about 1/3 of total luminal cells. Looking at the immunostaining images, it appears that there is an overlap in the localization of SCGB1A1 and FOXJ1. One would not expect this as they are localized in different compartments.

4) Hybrid cell state origin and markers: In Figure 3, the authors talked about identifying a novel cilia-secretory hybrid cell state using *Gp2* and *Prr18* as markers. what are the different cell types express these markers? Are these markers only expressed by hybrid cell but no other cell types? In Figure 5I, the schematic indicates that the hybrid cell state originates from ciliated cells. It is unclear on what basis the authors claim that this hybrid cells state originates from ciliated cells. No experimental evidence is provided for this claim.

5) In Figure 3I authors show the enrichment of cilia-secretory hybrid cells state in multiple airway diseases (COPD, Asthma and PCD) and stated a critical role of this novel cell state in airway function and pathogenesis. However, it is important to establish the presence of this population in disease states in situ to make such claims.

6) There is no clear link between Figure 1, Figure 2, and Figure 3. What is the link between *Tmem16a* to the mapping monogenic and complex trait disease-associated cell types of the airway? Is there a relationship between *Tmem16a* mutation and to the cilia-secretory hybrid cell state? Are there any changes in the number of cells cilia-secretory hybrid cell state in *Tmem16a* mutants?

[Editors' note: further revisions were suggested prior to acceptance, as described below.]

Thank you for resubmitting your work entitled "Chloride channels regulate differentiation and barrier functions of the mammalian airway" for further consideration by *eLife*. Your revised article has been evaluated by Edward Morrisey (Senior Editor) and a Reviewing Editor.

The manuscript has been improved but there are some remaining issues that need to be addressed before acceptance, as outlined below:

As reviewer 3 points out, the data utilizing the *Foxj1-cre^ERT2^:GFP* crossed to the *mTmG* reporter does not allow for accurate inducible lineage tracing given the two different GFP readouts. Please amend your manuscript to either remove this data or revise the conclusions to more accurately describe these results including the hybrid cell state.

*Reviewer #3:*

The authors addressed all the comments. However, I have one comment that needs to be clarified.

In the lineage tracing data, I was a little confused to see that the GFP signal appears in cytoplasm. Looking at the methods section, I noticed that the authors used *Foxj1-Cre^ER^:*GFP crossed with ROSA *mT/mG*. The use of double reporters (GFP and mGFP) driven by same promoter is not appropriate for cell lineage tracing. In addition, none of the cells show membrane localization of GFP in these images. Therefore, it is difficult to determine if hybrid cells originated from ciliated cells.

Throughout the manuscript (except Materials and methods section) and the response letter, the authors have not mentioned regarding the use of GFP and mGFP in the same mouse line. Previous studies used *FoxJ1-CRE* (not creER) and did not find any lineage labeled secretory cells (Pardo et al., 2013). Therefore, I suggest the authors either remove this data and the claims about hybrid cell state or clarify with appropriate mouse models/data.

---

## [Author Response]

Reviewer #1:[…]1) The authors report in Figure 1A, B that Tmem16a^-/-^ knockout results in mucus obstruction of the respiratory tract and alveolar simplification in neonatal mutants. What is provided is poorly documented and insufficient. The PAS staining is of poorly quality to demonstrate mucus obstruction; the goblet cell metaplasia needs to be further documented by Alcian Blue and Muc5a stainings.

In this revision, we have included high-resolution images of Alcian Blue stains to demonstrate mucus obstruction (Figure 1B; Figure 1—figure supplement 1B-E). We have also included immunostainings using an SCGB1A1 antibody to demonstrate the expansion of the secretory cell population in newborn airway in *Tmem16a^-/-^* mutants (Figure 1—figure supplement 1A).

Based on immunostaining of SCGB1A1, a typical secretory cell marker, and single cell RNAseq analysis, we observed consistent expansions of the secretory cell population in the mutant airway. We do not claim that this *Tmem16a*-dependent expansion of secretory cells is a typical goblet hyperplasia for the following reasons. First, at the current resolution of our single cell analysis, we have not observed different clusters of secretory cells that are separated by previously reported club cell markers or by goblet cell markers (Montoro et al., 2018). We therefore annotated that cluster as secretory cells rather than club cells or goblet cells. Second, in addition to SCGB1A1, differentially expressed markers for this secretory cell cluster in the wild-type newborn airway include *Gp2* and *Tff2*, both of which are molecular markers for adult goblet cells (Montoro et al., 2018). Therefore, our data indicate a clear temporal feature of secretory cell transcriptomes in the neonatal airway that is distinct from that of the adult club cells. Third, we analyzed *Muc5ac* mRNA expression in different types of newborn airway cells and did not observe any significant difference in the expression of Muc5ac in wild-type and *Tmem16a^-/-^* mutants, although we observed elevated expression of *Muc5b* and *Muc*4 in secretory cells in the mutants. Author response image 1 shows the expression levels of *Muc5ac* in P1 and P4 wild-type and mutant airway epithelial cells:

**Author response image 1. respfig1:** Muc5ac expression.

The region depicted by the square (I 2B WT), looks rather a blood vessel and is very different from the airway shown I 2B in the mutant lung. Moreover, from the picture provided in Figure 1B there is no convincing evidence of alveolar simplification in mutant, which should be shown by morphometrical analysis. It is unclear whether the authors mean to report a defect in sacculation at P1. The alveolar phenotype should be confirmed at P4 or later, since alveolar formation initiates at around P3. How do the authors explain this phenotype? Is Tmem16a expressed in the alveolar compartment?

We now provide additional documentation of lung phenotypes in Figure 1—figure supplement 1. Because *Tmem16a* mutants are perinatally lethal by the first week of birth, we normally analyze the airway phenotypes between P1 to P4. In Figure 1B, we analyzed airways from P3 control mice and mutant mice. In Figure 1—figure supplement 1D-E, we included samples from P5 control and mutants. In both P3 and P5 lung sections, the air sac space is consistently larger in the mutants. *Tmem16a* is only sparsely expressed in the alveolar space (Tabula Muris/PMID: 30283141/https://tabula-muris.ds.czbiohub.org/). We speculate that the defects may be secondary to inflammation associated with an abnormal barrier function in the neonatal airway.

2) What is the significance of the reversed mucociliary flow (0.42 ± 0.18 μm/s) in Tmem16a mutants and how could this be mechanistically explained and associated with the phenotype?

Defective mucociliary clearance has been observed in CF animal models (Hoegger et al., 2014) and asthma patient samples. We speculate the reduced flow and altered flow patterns may be related to increased secretory cells, abnormal mucus structure, deficiencies in anion secretion, and potentially tracheomalacia. A reduced flow will lead to longer airway clearance time and may exacerbate mucus obstruction observed in *Tmem16a* mutants.

3) There is insufficient information about how the parameters depicted in Figure 2 were analyzed and displayed. For example, in Figure 2B the heatmap shows average expression of marker genes for each cluster identified at E15, E16, P1 and P4. However, there is no indication of how gene expression at the ages indicated are displayed. Methods and result sections are vague and uninformative. Same for the human samples in Figure 2C.

For mouse airway atlas, we included marker gene expressions for all cell states and types identified from E15 to P4. Temporal distributions of cell states and types is shown in Figure 2—figure supplement 1B. Expression for marker genes are normalized to all cells included in this analysis. For human samples, we observed similar cell types and expression profiles for all cells collected from gestation week 21 and 23. We have provided more experimental details in the revised manuscript.

4) There is no lineage study or other evidence to support the authors' statement that the hybrid ciliated-secretory cells derive from ciliated cells. There are no definitive follow up experiments to determine whether these cells represent a transient state and whether they originate from secretory or ciliated cells. As described, the analysis of Pofut1^-/-^ in Figure 4—figure supplement 1 neither convincingly demonstrate the existence of these hybrids cells, nor identify their origin. The Arl13b-mCheery/Centrin-GFP reporter mice are mentioned in the Materials and methods section but data are not shown or discussed.

In this revised version of our manuscript, we have performed lineage tracing experiments using *Foxj1Cre^ERT2^*to demonstrate that the ciliated cell lineage is a major source for cilia-secretory hybrid cells in the neonatal airway. We set up crosses using *Foxj1Cre^ERT:GFP^*mice to *Rosa26mT/mG* reporter mice and administered Tamoxifen at E14 to E15 to induce the expression of *Rosa mGFP* in ciliated cells at the onset of airway differentiation. We then analyzed whether some of these labeled cells express the secretory cell marker SCGB1A1 at P0 to P1 using immunostainings. About half of the *GFP* labeled cells show cytoplasmic expression of SCGB1A1, consistent with our scRNA seq profiling of the airway cell types in the newborns. These results are now included in Figure 5E. We believe this hybrid cell state is a transient cell state given that in our dataset this hybrid cell state peaks at newborns and gradually decreases postnatally. In addition, the hybrid cell population has not been observed in adults based on previously published work (Montoro et al., 2018; Plasschaert et al., 2018).

We crossed *Tmem16^+/-^* with Arl13b-mCherry/Centrin-GFP reporter mice to better quantify ciliated cells based on the expression of Centrin-GFP. Images of Centrin-GFP expression in the airway of wild-type and mutants are now included in Figure 1—figure supplement 1A.

5) Spdef and Creb3l1 are not involved in the specification of embryonic (E16) secretory cells. In fact, Spdef expression is reported to initiate later in the developing lung and postnatally. Moreover, Spdef is not associated with induction of secretory (Club). In fact, Spdef inhibits Club cell differentiation and induces the goblet cell program (Park et al., 2007).

We thank the reviewer for pointing out that *Spdef* is shown to promote goblet cell hyperplasia. In the absence of *Spdef*, goblet cells fail to develop. Our data show that the embryonic secretory cells express *Spdef* at E16 (Figure 3B, D). In addition, neonatal secretory cells exhibit a molecular signature similar to those of adult goblet cells rather than to those of club cells. For example, neonatal secretory cells express *Gp2* and *Tff2*, both are goblet cell markers of adult mouse trachea (Montoro et al., 2018). The data suggest a potentially conserved regulatory program for secretory cells in different developmental stages and in different tissues. We have revised our text to avoid confusion around the requirement of *Spdef* or *Creb3l1* for the specification of secretory cells. We only use these marker genes to indicate the gene modules for secretory program are upregulated at E16.

6) It is not clear what the authors mean by "By demonstrating the similarities in airway defects as well as in expression patterns of Tmem16a and CFTR in mouse and human airway epithelium, respectively, our work provides an entry point for understanding the developmental aspects of airway barrier pathogenesis". In spite of the focus on chloride channels and comparisons with CFTR, at the end it unclear what Tmem16a is really doing in these cells.

We demonstrate in this study that *Tmem16a* in the undifferentiated epithelial cells plays an essential role in maintaining airway progenitors by limiting the differentiation of basal cells into the secretory lineage. Because *Tmem16a* and *CFTR* are expressed in orthologous cell types in the developing mouse and human airways, respectively, and because removal of *Tmem16a* from the mouse airway recapitulate key aspects of CF symptoms, we believe our work presents a tractable mouse model that allows for discoveries of airway cell types that require chloride channels for proper differentiation and functions that are relevant to early onset airway diseases.

Reviewer #2:[…]1) Figure 1A showed that Tmem16a null mutant mice had increased mucus secretion. Are the levels of submucosal glands and goblet cells increased in the mutant?

We thank the reviewer for the comments. In this version of the manuscript, we have included high-resolution images of Alcian Blue stains to demonstrate mucus obstruction (Figure 1B; Figure 1—figure supplement 1B,C,D,E). We have also included immunostainings using SCGB1A1 to demonstrate the expansion of secretory cell population in newborn airway in *Tmem16a^-/-^* mutants (Figure 1—figure supplement 1A).

Submucosal glands are largely absent from the main segment of the newborn airway (cartilage ring 1-12) and we did not observe these cells in our single-cell analysis.

We do not claim that this *Tmem16a*-dependent expansion of secretory cells is a typical goblet hyperplasia for the following reasons: First, at the current resolution of our single cell analysis, we have not observed different clusters of secretory cells separated by previously reported club cell markers or by goblet cell markers (Montoro et al., 2018). We therefore annotated that cluster as secretory cells rather than club cells (the major secretory cells). Second, in addition to SCGB1A1, differentially expressed markers for this secretory cell cluster in the wild-type newborn airway include *Gp2* and *Tff2*, both of which are molecular markers for adult goblet cells (Montoro et al., 2018). Therefore, our data indicate a clear temporal feature of secretory cell transcriptomes in the neonatal airway that is distinct from that of the adult club cells. Third, we analyzed the expression of *Muc5ac*, a typical marker for adult goblet cells, in different types of newborn airway cells. We did not observe any significant difference in the expression of *Muc5ac* in wild-type and *Tmem16a^-/-^* mutants. Although we observed elevated expression of *Muc5b* and *Muc*4 in secretory cells in the mutants.

2) Figure 1E showed increased number of SCGB1A1^+^ cells in Tmem16a null mutants. It is unclear whether Tmem16a impacts cell proliferation. Does Tmem16a deletion increase SCGB1A1^+^ cell proliferation?

No change in epithelial proliferation in *Tmem16a* null mutants was previously observed by Rock et al., 2008. In this revised manuscript, we have analyzed the cell cycle scores for different cell states at E16 airway during airway differentiation. By analyzing the expression levels of a large number of cell-cycle related genes, we did not observe any significant change in cell cycle scores between wild-type and mutant secretory cells. The results are included in Figure 6—figure supplement 1B.

3) Figure 6A, B showed Tmem16a expression in E15 mouse lung epithelium. It is unclear whether Tmem16a is expressed in postnatal lung epithelium during both homeostasis and Polidocanol-induced airway regeneration.

We apologize for the confusion. We have included additional data in Figure 7—figure supplement 1 to demonstrate that *Tmem16a* is expressed in the adult airway, primarily in secretory cells and basal cells, in both mice and humans.

4) Figure 4C showed that Tmem16a null mutant epithelial cells significantly downregulated NF-κB inhibitors Nfkbia and Nkfbiz. It is unclear whether decreased Nfkbia and Nkfbiz globally impacts NF-κB signaling. Does Tmem16a deletion increase NF-κB nuclear activity in lung epithelial cells?

The NF-κBpathway outcome appears to be more complex depending on the stage and tissue types. In the current manuscript we have thus revised the text to avoid confusion and overstatement.

5) Tmem16a null mutants displayed an expansion of immature secretory cell populations. I wonder if this phenotype was caused directly by the loss of Tmem16a or indirectly via increased inflammation. Does reduced inflammation recue the phenotype observe in the Tmem16a null mutant?

The reviewer is indeed correct that sustained inflammation postnatally can induce goblet cell hyperplasia. In our analysis, these immature secretory cells that are accumulated in the *Tmem16a* null mutants appeared at E16. At that stage, many immune modulates are either not expressed or expressed at comparable levels between control and mutants (Figure 5—figure supplement 1B). The data thus suggest that the early expansion of secretory cells in *Tmem16a* mutants is independent of an inflammatory response.

6) Figure 6H showed that deletion of Tmem16a in adult Krt5^+^ cells resulted in decreased ratio of ciliated cells versus secretory cells during Polidocanol-induced airway regeneration. Do the Krt5-creERT2, Tmem16a fl/fl mice exhibit similar phenotypes as those observed in Tmem16a null mutants, such as mucus obstruction and increased inflammation?

We have not systematically analyzed the airway phenotypes of *Tmem16a* conditional KOs. Given that neonatal and adult immune systems can be quite different, it is likely that the inflammatory response in the *Tmem16a* conditional KOs (adults) is manifested in different ways from the *Tmem16a* null mutants (neonates). We have analyzed n = 2 *Tmem16a* conditional KO mice 4 weeks after polidocanol induced injury. Compared to control mice, these conditional KO mice showed normal appearance and weight. Histology analysis of the airway reveals no obvious mucus obstruction but shows infiltrated immune cells and cell debris in the airway lumen in the conditional mutants. Future mechanistic studies will be required to determine the role of *Tmem16a* in generating barrier immune response during the regenerative phase.

7) Figure 6G showed that deletion of Tmem16a in mouse epithelium led to mucous metaplasia and biased differentiation toward secretory cells during postnatal life. It's interesting that prior studies have indicated that Tmem16a overexpression led to mucous hyperplasia and blocking Tmem16a suppressed mucin secretion in the lung secretory epithelial cell (Huang et al., 2012). These studies suggest that both loss- and gain-of-expression of Tmem16a in lung epithelial cells could cause mucous hyperplasia. It is unclear about the reasons underlying these contradictory findings. The rationale and discussion are needed to clarify these findings.

We thank the reviewer for this thoughtful comment. TMEM16A overexpression indeed has been observed in asthma-like epithelial cells. However, there is no indication that overexpression of TMEM16A can induce mucus overproduction in vivo or in vitro. There was a discussion on this topic at the European CF meeting a few months ago, which is nicely summarized in the following article Amaral and Beekman, 2019.

Reviewer #3:In this manuscript, He et al., studied the role of chloride channel Ano1/Tmem16a in airway development and its role in barrier function. The authors used Tmem16 knockout mouse model and claim that inactivation of Tmem16 results in alveolar simplification and obstruction of the respiratory tract due to mucus accumulation in neonatal mutants. To characterize the cellular origins of airway defects, the authors performed single-cell RNA sequencing of mouse trachea on wild-type and Tmem16a mutant cells from embryonic tissues. Further the authors profiled human fetal tracheal cells at gestation weeks 21 and 23 (GW21 and GW23) to identify orthologous cell types and cell states defined by similar molecular markers between the mouse airway and the human airway. Based on this data, the authors claim to have identified a novel cilia-secretory hybrid sell state using Foxj1, Gp2 and Prr18 as markers. Additionally, they state that these hybrid cells are likely derived from a ciliated cell lineage and play a critical role in airway function and pathogenesis including Asthma, COPD, and PCD. Finally, using scRNAseq the authors claim that loss Tmem16a results in secretory cell hyperplasia and Tmem16a inhibits basal progenitor differentiation towards the secretory lineage.Overall, the authors used scRNA-sequencing to understand the development of airway epithelium and to study the role of Tmem16 in this process. However, the data presented here is incoherent and lacks clear message. The data provided in the current manuscript is of poor quality and does not add much to the knowledge in this area. It appears to me that the authors put together two unrelated pieces (scRNA-seq and Tmem16). The authors point to a hybrid cell state that's representative of both ciliated and secretory cells. As pointed in the later section, the numbers and proportions of these intermediates is unclear, and the validation data presented here is not convincing. Similarly, the authors talked about genes enriched in CF, COPD, asthma and COPD but there is no direct correlation to any of the phenotypes described here.

We thank the reviewer for the comment. Through a combination of single cell analyses and in vivo characterizations of mouse mutants, we have uncovered genetic signatures of airway development at various states of differentiation and identified the cellular origin of mucus cell hyperplasia in mouse mutants that lack *Tmem16a*. TMEM16A is a candidate drug target in the modulation and management of cystic fibrosis. Despite many efforts to identify agonists and activators for TMEM16A, the physiological functions for TMEM16A in the mammalian airway remained unclear. Single-cell analysis in our study proves to be a very powerful approach to demonstrate that TMEM16A acts in the airway basal progenitor cells and controls critical steps of airway differentiation. We believe these critical roles of *Tmem16a* define an entirely new aspect of chloride channel biology and provide new insights into the developmental and cellular origins of early onset airway diseases.

Essential revisions:1) Tmem16 mutant characterization: In Figure 1, the authors present data from Tmem16 mutant airway tissue characterization. The authors compared the muco-ciliary clearance between the wild-type and Tmem16a^-/-^ knockout mouse line without describing any experimental details. In Figure 1A, the authors used Jacalin-488 staining. However, it is unclear why the authors chose to use Jacalin as this is not a commonly used marker for airway cell types. From the images, it appears that it marks all luminal cells of the airway epithelium. Co-staining with secretory and ciliated cell markers would help.

We thank the reviewer for the thoughtful comments. We have provided more documentations of airway phenotypes in the revised manuscript. We used Jacalin Lectins to label mucin producing cells given it is able to bind to glycoproteins, which are major components of the airway mucus (Ostedgaard et al., 2017). We have also included immunostaining of SCGB1A1 in newborn airway to demonstrate the expansion of secretory cells in the mutants (Figure 1—figure supplement 1A). Our histology analysis is consistent with our scRNA profiling results.

For muco-ciliary clearance experiments, we have also included more experimental details for this section in the main text and method sections.

In Figure 1B, PAS staining does not look convincing – a better representative image can be included.

In this revision, we have provided more documentations of lung phenotypes (Figure 1—figure supplement 1). Because *Tmem16a* mutants are perinatally lethal by the first week of birth, we normally analyze the airway phenotypes between P1 to P4. In Figure 1B, we analyzed airways from P3 control and mutants. In Figure 1—figure supplement 1D, E, we included samples from P5 control and mutants. In both P3 and P5 airway sections, we consistently observed mucus obstruction in the mutants.

Further, authors talked about defects in mucociliary clearance (Figure 2C) by assessing the flow dynamics of fluorescent beads in Tmem16a^-/-^ mouse without describing the tracheal abnormalities in Tmem16a^-/-^ knockout mouse? What are the anatomical and structural differences between wild-type trachea to Tmem16a^-/-^ knockout? Previous studies have shown that the tracheal cartilage rings are abnormal in the mutants. So, is it possible that the MCT defects observed here are due to abnormalities in tissue structure? And also, some of the phenotypes shown here have been already described elsewhere. The authors failed to cite the previous studies in this section.

Defective mucociliary clearance has been observed in CF animal models (Hoegger et al., 2014) and asthma patient samples. We speculate the reduced flow and altered flow patterns may be related to increased secretory cells, abnormal mucus structure, deficiencies in anion secretion, and potentially tracheomalacia. Further studies will be required to dissect how each aspect of airway abnormality can contribute to mucociliary flow defects. A reduced flow will lead to longer airway clearance time and may exacerbate mucus obstruction observed in *Tmem16a* mutants. We have included additional reference for this section.

2) Correlation between Tmem16 expression and function: In the current study authors used embryonic and neonatal mouse tissue and studied the role of Tmem16a in formation and function of the airway mucosal barrier without clearly characterizing the Tmem16a expression pattern at different developmental stages. Authors state that Tmem16a is expressed in undifferentiated mouse epithelium as well as in differentiated secretory cells at E15 (Figure 6A-B). However, In the Figure 2E authors shows their single cell RNA-seq identify Ano1/Tmem16a is specific to the mouse secretory cells. Does that mean later in development or in neonates only secretory cells express Ano1/Tmem16a? if so, at which developmental stage Ano1/Tmem16a expression is specified to the secretory cells?

We apologize for the confusion. We have now included detailed characterization of *Tmem16a* expression in different cell types during development (Figure 6) and homeostasis (Figure 7—figure supplement 1). *Tmem16a* is indeed expressed in undifferentiated basal progenitor cells at E15 and become enriched in the secretory cell lineage upon differentiation. Low levels of *Tmem16a* expressed is maintained in basal cells at both developmental and homeostasis phases, and the expression pattern is conserved in mouse and human airway (Figure 7—figure supplement 1).

3) scRNA-seq data and hybrid cell state: The authors performed single-cell RNA sequencing of mouse trachea from wild-type embryonic day 15 (E15), E16, P1, and P4 and Tmem16a mutant trachea of E16, P1, and P4 to characterize the cellular origins of these complex airway defects. However, they did not show as to how does the comparative data look like? What are the major transcriptomic differences between the wild-type and Tmem16 mutant trachea? How does the cellular landscape look like on tSNE/UMAP?

We thank the reviewer for this comment. To clarify, we have provided many detailed characterizations to demonstrate how *Tmem16a* mutants differ from wild-type littermates. For example, we show that cellular composition of airway epithelial cells are different between wild-type and mutants (Figure 6—figure supplement 1A; Figure 7B); we show that mucosal barrier genes are differentially expressed in wild-type and mutants (Figure 5C); we show that a specific intermediate state of secretory cells is expanded in the mutants during airway differentiation at E16 (Figure 7B). The *tSNE* plots shown below are separated by genotypes: mutant in red and wild-type in blue, and arranged by age, E16 (left) to P1 (middle) to P4 (right). Transcriptomic differences become more apparent as the animals develop. We have characterized the altered basal-to-secretory cell differentiation program in *Tmem16a* mutants at E16. In the revised manuscript, we also include the transcriptomic differences of the P4 secretory cells in Figure 5—figure supplement 1A.

**Author response image 2. respfig2:** tSNE plots of single cell transcriptomes for control and Tmem16a KO mutant tracheal cells at E16, P1, and P4.

Further, they assign the percent distribution of each cell population based on scRNA-seq data (Figure 2—figure supplement 1B). Does that represent the actual distribution of these cell populations in mouse trachea? Did authors account for any artifacts associated with tissue dissociation and sampling?

We thank the reviewers for the comment. The cell type profile we presented here may not reflect the actual numbers for all cell types, but we have included many cells in our sampling, and the distribution profile is supported by previous literatures including both in vivo characterization and single-cell analyses. Most importantly we compare the cellular composition across different time points and genotypes. Given the comparative nature of our analyses pertaining to the cellular composition, we are confident that the differences we see are true since technical variations are consistent and maintained for all different conditions. We have included more discussion pertaining to cell composition and technical variation in Figure 2—figure supplement 2 legend.

Immunofluorescence data from Figure 3F and G indicates that the hybrid population accounts for about 1/3 of total luminal cells. Looking at the immunostaining images, it appears that there is an overlap in the localization of SCGB1A1 and FOXJ1. One would not expect this as they are localized in different compartments.

In this revised manuscript, we have performed lineage tracing experiments using *Foxj1Cre^ERT^*to demonstrate that the ciliated cell lineage is a major source for cilia-secretory hybrid cells in the neonatal airway. About half of the *GFP* labeled cells show cytoplasmic expression of SCGB1A1, consistent with our scRNA seq profiling of the airway cell types in the newborns. The results are now included in Figure 5E. We indeed observed in some cases that FOXJ1 staining is outside of the nucleus. Changes in subcellular localization may represent a mechanism to turn off FOXJ1-dependnent transcription.

4) Hybrid cell state origin and markers: In Figure 3, the authors talked about identifying a novel cilia-secretory hybrid cell state using Gp2 and Prr18 as markers. what are the different cell types express these markers? Are these markers only expressed by hybrid cell but no other cell types? In Figure 5I, the schematic indicates that the hybrid cell state originates from ciliated cells. It is unclear on what basis the authors claim that this hybrid cells state originates from ciliated cells. No experimental evidence is provided for this claim.

Based on our scRNA-seq analysis and in situ validation, *Gp2* is a secretory cell marker and *Prr18* is a ciliated cell marker (Figure 4A, B). *Gp2* is expressed in secretory cells and hybrid cells, while *Prr18* is expressed in ciliated cells as well as hybrid cells. In this revised manuscript, we have performed lineage tracing experiments using *Foxj1Cre^ERT^*to demonstrate that the ciliated cell lineage is a major source for cilia-secretory hybrid cells in the neonatal airway. We set up timed breeding with *Foxj1Cre^ERT^*mice and administered Tamoxifen at E14 to E15 to induce the expression of *Foxj1-GFP* in ciliated cells at the onset of airway differentiation. We then analyzed whether some of these labeled cells express secretory cell marker SCGB1A1 at P0 to P1 by immunostainings. About half of the *GFP* labeled cells show cytoplasmic expression of SCGB1A1, consistent with our scRNA-seq profiling of the airway cell types in the newborns. The results are now included in Figure 5E. We believe this hybrid cell state is a transient cell state given that it appears in our dataset this hybrid cell state peaks at newborns and gradually decrease postnatally and has not been observed in adults based on previously published work.

5) In Figure 3I authors show the enrichment of cilia-secretory hybrid cells state in multiple airway diseases (COPD, Asthma and PCD) and stated a critical role of this novel cell state in airway function and pathogenesis. However, it is important to establish the presence of this population in disease states in situ to make such claims.

We apologize for the confusion. We developed a single-cell Gene Percentile Score analysis to characterize cell-type specific expression patterns of airway disease associated genes. The information we want to convey in previous Figure 3l (now in Figure 4F of the revised manuscript) is that the disease-associated genes (please refer to Figure 4—figure supplement table 1 for the full list of genes reported to be associated with each disease) are highly expressed in cilia-secretory hybrid cells. We therefore suggest that these cells may be involved in the pathogenesis of various types of airway diseases. Two independent studies indeed demonstrated that cilia-secretory hybrid cells are present in human patients with asthma and choric allergy and absent from healthy controls (Vieira Braga et al., 2019, Ordovas-Montanes et al., 2018). Earlier studies (Turner et al., 2011 and Tyner et al., 2006) using cultured human airway epithelial cells showed that *Foxj1*-lineage cells can contribute to goblet cell hyperplasia in cultured human airway epithelia cells upon virus infection or IL-13 treatment. These lines of evidence suggest that the emergence of hybrid cells may depend on inflammatory signals. We have provided additional explanation and discussion in the revised manuscript.

6) There is no clear link between Figure 1, Figure 2, and Figure 3. What is the link between Tmem16a to the mapping monogenic and complex trait disease-associated cell types of the airway? Is there a relationship between Tmem16a mutation and to the cilia-secretory hybrid cell state? Are there any changes in the number of cells cilia-secretory hybrid cell state in Tmem16a mutants?

We thank the reviewer for the thoughtful comment. *Tmem16a* mutants show complex airway defects and can be used as a tractable mouse model to understand aspects of human airway diseases. scRNA seq combined with mouse genetics and in vivo characterization affords the opportunity to systematically and unbiasedly identify the cellular origin of airway defects in the absence of *Tmem16a* and to generate hypotheses for future mechanistic studies. In our analysis, there is no changes in the number of hybrid cells. However, hybrid cells express many genes related to barrier function (Figure 5A), and the expression levels of these barrier genes are significantly altered in *Tmem16a* mutants (Figure 5C).

[Editors' note: further revisions were suggested prior to acceptance, as described below.]

Reviewer #3:The authors addressed all the comments. However, I have one comment that needs to be clarified.In the lineage tracing data, I was a little confused to see that the GFP signal appears in cytoplasm. Looking at the methods section, I noticed that the authors used Foxj1-CreER-GFP crossed with ROSA mT/mG. The use of double reporters (GFP and mGFP) driven by same promoter is not appropriate for cell lineage tracing. In addition, none of the cells show membrane localization of GFP in these images. Therefore, it is difficult to determine if hybrid cells originated from ciliated cells.Throughout the manuscript (except Materials and methods section) and the response letter, the authors have not mentioned regarding the use of GFP and mGFP in the same mouse line. Previous studies used FoxJ1-CRE (not creER) and did not find any lineage labeled secretory cells (Pardo et al., 2013). Therefore, I suggest the authors either remove this data and the claims about hybrid cell state or clarify with appropriate mouse models/data.

We agree with reviewer 3 that using double reporters GFP (*Foxj1-CreER-GFP*) and mGFP (*ROSA mT/mG*) may not be ideal. However, the endogenous GFP fluorescence signal from *Foxj1-CreERT2-GFP* is very weak. Given previous examples that used *ShhCre-GFP* and *ROSA mT/mG* to trace ShH^+^ lineage (Kuo and Krasnow, 2015) or used *Prx1-CreERT2-EGFP* and ROSA mT/mG to trace Prx1^+^ lineage (de Lageneste et al., 2018), we crossed this *Foxj1-CreERT2-GFP* mouse line with *ROSA mT/mG* to introduce mGFP reporter for lineage tracing. Before fixation, we isolated trachea samples under a fluorescence dissection microscope and only processed those with both Tomato and GFP signals. After fixation and whole-mount staining, samples were flat mounted on 35 mm No 1.5 MatTek dishes (luminal side facing the coverslip) and imaged *en-face*. For immunostaining, mGFP signal was used to indicate Foxj1^+^ lineage. mTomato signal was quenched after a sequential fixation by PFA and methanol, so we used E-cad antibody to label cell membrane in the RFP channel and pseudo-colored it in grey in the image panel. A mouse monoclonal antibody for SCGB1A1 was used to label secretory cells in the far-red channel. The images we showed in the figure were taken from an entire 3D projection, including the apical surface. It may look as if there is no visible localization of membrane GFP, but mGFP is clearly visible in the sub-apical sections of the trachea sample. We have revised the figure (Now Figure 4—figure supplement 1B, C) to that majority of the GFP labeled cells in our analysis expressed mGFP. We also included more description in the revised text, figure legend, and methods to more accurately describe this experiment.

Regarding the evidence for hybrid cells, we have provided validation of their presence using orthogonal methods (single-cell RNA sequencing, RNA FISH analysis, and TEM) in our manuscript, and we believe that the identification of those hybrid cells will be an important addition to airway biology. Similarly, cilia-secretory hybrid cells have been observed in nasal and airway epithelial tissue samples in patients with allergy and asthma (Ordovas-Montanes et al., 2018 and Vieira Braga et al., 2019). It will be very interesting to determine whether neonatal hybrids cells reported in our study and asthma-associated hybrids cells from human patients have similar cellular origins, which will be defined by different and complementary lineage tracing approaches.

Reviewer 3 pointed out that in Pardo et al., 2013, *FoxJ1-CRE* (not creER) did not label any lineage labeled secretory cells upon challenge of ovalbumin in adult mice. On the other hand, Turner et al., 2011 and Tyner et al., 2006 both showed that *Foxj1*-lineage cells can contribute to goblet cell hyperplasia in cultured human airway epithelia cells upon IL-13 treatment. More evidence supporting a cilia-to-goblet trans-differentiation was reviewed by Patel et al., 2011 and further supported by Vieira Braga et al., 2019. The exact lineage for cilia-secretory trans-differentiation may depend on different physiological and pathological conditions. For example, it may depend on age (i.e., neonatal vs adult), on the types of immune responses (i.e., acute injury vs allergy), on the types of stimuli (i.e., ovalbumin, dust mite, or more complex causes for asthma in patients), and on the duration of allergen challenges (i.e., multiple treatments of ovalbumin used in Pardo et al., 2013, 14 days of IL-13 treatment used in Turner et al., 2011, or years of chronic inflammation in asthma patients reported by Vieira Braga et al., 2019). Future studies will be required to determine how hybrid cells can be induced by distinct intrinsic and extrinsic signals, which are beyond the scope of our current manuscript.